# The changing role of organic nitrates in the removal and transport of $NO_x$

Paul S. Romer Present[1], Azimeh Zare[1], and Ronald C. Cohen[1,2]

[1]Department of Chemistry, University of California Berkeley, Berkeley, CA, 94720, USA.
[2]Department of Earth and Planetary Sciences, University of California Berkeley, Berkeley, CA, 94720, USA.

**Correspondence:** Ronald C. Cohen (rccohen@berkeley.edu)

**Abstract.** A better understanding of the chemistry of nitrogen oxides ($NO_x$) is crucial to effectively reducing air pollution and predicting future air quality. The response of $NO_x$ lifetime to perturbations in emissions or in the climate system is set in large part by whether $NO_x$ loss occurs primarily by the direct formation of $HNO_3$ or through the formation of alkyl and multifunctional nitrates ($RONO_2$). Using 15 years of detailed in situ observations, we show that in the summer daytime continental boundary layer the relative importance of these two pathways can be well approximated by the relative likelihood that OH will react with $NO_2$ or instead with a volatile organic compound (VOC). Over the past decades, changes in anthropogenic emissions of both $NO_x$ and VOCs have led to a significant increase in the overall importance of $RONO_2$ chemistry to $NO_x$ loss. We find that this shift is associated with a decreased effectiveness of $NO_x$ emissions reductions on ozone production in polluted areas and increased transport of $NO_x$ from source to receptor regions. This change in chemistry, combined with changes in the spatial pattern of $NO_x$ emissions, is observed to be leading to a flatter distribution of $NO_2$ across the United States, potentially transforming ozone air pollution from a local issue into a regional one.

## 1 Introduction

Nitrogen oxides ($NO_x \equiv NO + NO_2$) play a central role in the formation of toxic air pollutants including $O_3$ and secondary aerosols. More broadly, $NO_x$ chemistry controls the rates and pathways of atmospheric oxidation by determining the concentration of the three most important tropospheric oxidants: OH, $O_3$, and $NO_3$. $NO_x$ emissions also directly contribute to nitrogen deposition in sensitive ecosystems (Fowler et al., 2013). Due to its harmful effects to the environment and human health, $NO_x$ has been the target of emissions control strategies since the 1970s, causing anthropogenic $NO_x$ emissions in the United States to have decreased by a factor of 2 or more over the past 30 years (United States Environmental Protection Agency, 2018). Understanding the consequences of these past changes and predicting the results of future emissions reductions on the atmosphere requires a quantitative description of feedbacks between $NO_x$ concentrations and $NO_x$ chemistry.

After emission to the atmosphere, removal of $NO_x$ occurs through two primary pathways: conversion to $HNO_3$ and conversion to alkyl and multifunctional nitrates ($RONO_2$). Once formed, $HNO_3$ is nearly chemically inert in the troposphere, with a lifetime to reaction or photolysis of over 50 hours. $HNO_3$ is therefore removed almost entirely by wet and dry deposition. $RONO_2$ represents a class of diverse molecules, with atmospheric lifetimes ranging from hours to days depending on

the properties of the organic backbone (R-group). The loss of $RONO_2$ is divided among reactions that release $NO_x$ from the R-group and recycle it back to the atmosphere, reactions that result in heterogeneous hydrolysis to form $HNO_3$, and direct deposition. The latter two pathways permanently remove $NO_x$ from the atmosphere (Nguyen et al., 2015; Romer et al., 2016; Fisher et al., 2016). Other $NO_x$ oxidation products, such as peroxy acetyl nitrate (PAN) or HONO can play an important role

in the transport and redistribution of $NO_x$ but do not generally lead to permanent $NO_x$ removal.

    Historically, direct $HNO_3$ production was thought to be the only important $NO_x$ loss pathway, with $RONO_2$ chemistry playing at most a minor role. However, several studies have shown that the formation rate of $RONO_2$ in cities or forested regions can be competitive with or greater than the direct production rate of nitric acid (Rosen et al., 2004; Farmer et al., 2011; Browne et al., 2013; Romer et al., 2016; Sobanski et al., 2017).

The relative importance of $HNO_3$ and $RONO_2$ production is an important factor in setting the lifetime of $NO_x$ (Romer et al., 2016) and affects the response of $NO_x$ loss to temperature (Romer et al., 2018). Due to their different production pathways, the relative importance of $HNO_3$ and $RONO_2$ production also controls how $NO_x$ loss and ozone production are affected by changes to emissions of $NO_x$ or volatile organic compounds (VOCs). By terminating the radical chain reactions, the formation of $RONO_2$ serves to suppress ozone formation in polluted areas (Perring et al., 2010; Farmer et al., 2011; Edwards et al., 2013;

Lee et al., 2014). Several studies have also shown that $RONO_2$ can efficiently partition into aerosols, potentially explaining a large portion of secondary organic aerosol in a wide range of environments (Rollins et al., 2012; Pye et al., 2015; Xu et al., 2015b; Lee et al., 2016).

    Multiple previous studies have used chemical transport models to investigate how the relative production of $RONO_2$ and $HNO_3$ varies in different environments. Browne and Cohen (2012) modeled $NO_x$ loss over the Canadian boreal forest using

WRF-Chem and Fisher et al. (2016) and Zare et al. (2018) studied $NO_x$ loss in the southeast United States using GEOS-Chem and WRF-Chem respectively. These studies agree that in rural and forested areas with lower $NO_x$ emissions and higher biogenic VOC emissions, $RONO_2$ chemistry is often the largest sink of $NO_x$.

    However, these studies diverge in their conclusions about the overall importance of $RONO_2$ chemistry as a $NO_x$ sink and how it is likely to change in the future. In a WRF-Chem simulation identical to those described in Zare et al. (2018), $RONO_2$

chemistry is found to be 60 % or more of the total $NO_x$ loss across broad swathes of the southeast United States (Fig. 1), while Fisher et al. (2016) found $RONO_2$ production to be concentrated in rather small sections of the southeast. Furthermore, Fisher et al. (2016) suggested that the contribution of $RONO_2$ chemistry to $NO_x$ loss across the region is unlikely to change significantly in the future due to the spatial segregation of $NO_x$ and VOC emissions. On the other hand, Zare et al. (2018) and Browne and Cohen (2012) suggested that the contribution of $RONO_2$ chemistry to $NO_x$ loss was likely to grow significantly

if anthropogenic $NO_x$ emissions decreased across the United States.

    Here we use in situ observations from a collection of 13 different field deployments to investigate how the relative daytime production of $RONO_2$ and $HNO_3$ varies across the United States and how this fraction may change in the future. We show that the relative production of $RONO_2$ and $HNO_3$ can be well described by the relative OH reactivity of $NO_2$ and of the combined VOC mixture. As both anthropogenic $NO_x$ and anthropogenic VOC emissions have decreased substantially in the

United States over the past 20 years, the relative role of these two pathways has shifted as well. While the shift has generally

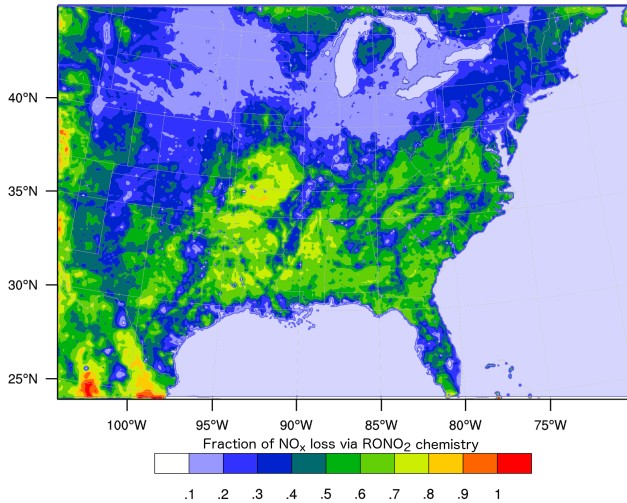

**Figure 1.** Average (24-hour) fraction of total $NO_x$ loss via $RONO_2$ chemistry over the southeast United States in summer 2013 simulated using the RACM2_Berkeley2 mechanism in WRF-Chem (Zare et al., 2018).

been towards an increasing role for $RONO_2$ chemistry, the shift has been smallest in large cities and largest in the transitional regime around them. Combined with changing emission patterns of $NO_x$, the shift in $NO_x$ chemistry is leading to a flatter distribution of $NO_x$ across the continental United States.

## 2   $NO_x$ chemistry and production of $RONO_2$ and $HNO_3$

$NO_x$ is emitted to the atmosphere as NO from a range of anthropogenic and biogenic sources, including motor vehicles, power plants, lightning, fires, and soil bacteria. In the daytime, NO interconverts with $NO_2$ on a timescale of minutes through Reactions (R1–R2), forming the chemical family $NO_x$. When $NO_x$ is combined with VOCs and hydrogen oxides ($HO_x$), a set of linked radical chain reactions is formed (R3–R6). As part of these reactions, two molecules of NO are oxidized to $NO_2$, leading to the net production of $O_3$ through Reaction (R2).

$$NO + O_3 \rightarrow NO_2 + O_2 \tag{R1}$$

$$NO_2 + h\nu + O_2 \rightarrow O_3 + NO \tag{R2}$$

$$OH + RH + O_2 \rightarrow RO_2 + H_2O \tag{R3}$$

$$\text{RO}_2 + \text{NO} \xrightarrow{1-\alpha} \text{RO} + \text{NO}_2 \tag{R4a}$$

$$\text{RO} + \text{O}_2 \rightarrow \text{R}'\text{CHO} + \text{HO}_2 \tag{R5}$$

$$\text{HO}_2 + \text{NO} \rightarrow \text{OH} + \text{NO}_2 \tag{R6}$$

The reactions that propagate the catalytic cycle occur at the same time as reactions that remove $\text{NO}_x$ from the atmosphere, terminating the cycle. Direct $\text{HNO}_3$ production occurs through the association of OH with $\text{NO}_2$ (R7). $\text{RONO}_2$ compounds are produced as a minor channel of the $\text{RO}_2 + \text{NO}$ reaction (R4b). Some fraction of the time $\alpha$, these two radicals will

associate to form an organic nitrate, with the balance forming $\text{NO}_2$ and eventually producing $\text{O}_3$ (R4a). The branching ratio $k_{\text{R4b}}/(k_{\text{R4a}} + k_{\text{R4b}})$ is designated $\alpha$ and is determined by the nature of the R-group as well as the temperature and pressure. Longer carbon backbones and lower temperatures increase $\alpha$, while lower pressures and oxygenated functional groups decrease it (Wennberg et al., 2018). Typical values of $\alpha$ in the summertime continental boundary layer range from near 0 for small hydrocarbons and highly oxygenated compounds to over 0.20 for large alkanes and alkenes (Perring et al., 2013).

$$\text{OH} + \text{NO}_2 + M \rightarrow \text{HNO}_3 + M \tag{R7}$$

$$\text{RO}_2 + \text{NO} + M \xrightarrow{\alpha} \text{RONO}_2 + M \tag{R4b}$$

The total rate of $\text{RONO}_2$ production can be calculated from the properties of individual VOCs measured in the atmosphere

via Eq. (1). In Eq. (1), $Y_{\text{RO}_2{}_i}$ represents the yield of $\text{RO}_2$ radicals from VOC oxidation and $f_{\text{NO}_i}$ represents the fraction of those $\text{RO}_2$ radicals that react with NO instead of reacting with $\text{HO}_2$ or undergoing unimolecular isomerization (e.g., Teng et al., 2017). $f_{\text{NO}_i}$ is close to 1 under polluted or moderately-polluted conditions, but decreases as the concentration of $\text{NO}_x$ decreases.

$$P(\text{RONO}_2) = [\text{OH}] \sum_{R_i} [\text{R}_i] \cdot k_{\text{OH}+\text{R}_i} \cdot Y_{\text{RO}_2{}_i} \cdot f_{\text{NO}_i} \cdot \alpha_i \tag{1}$$

If the contributions from individual VOCs are summed and averaged, the total production of $\text{RONO}_2$ can also be calculated from the effective behavior of the VOC mixture via Eq. (2), where VOCR is the sum of all measured VOC concentrations weighted by their reaction rate with OH.

$$P(\text{RONO}_2) = [\text{OH}] \cdot \text{VOCR} \cdot Y_{\text{RO}_2{}_{\text{eff}}} \cdot f_{\text{NO}_{\text{eff}}} \cdot \alpha_{\text{eff}} \tag{2}$$

In a similar fashion, the production of $HNO_3$ can be calculated via Eq. (3), where NO2R is the $NO_2$ reactivity, or the concentration of $NO_2$ multiplied by $k_{OH+NO_2}$. At 298 K and 1 atm, 10 ppb of $NO_2$ is equivalent to an NO2R of 2.3 s$^{-1}$.

$$P(HNO_3) = [OH] \cdot [NO_2] \cdot k_{OH+NO_2} = [OH] \cdot NO2R \tag{3}$$

Total $NO_x$ loss is the sum of the conversion to $HNO_3$ and conversion to $RONO_2$. The fraction of $NO_x$ loss via $RONO_2$ production can be expressed analytically as Eq. (4).

$$\frac{P(RONO_2)}{P(RONO_2) + P(HNO_3)} = \left(1 + \frac{1}{\alpha_{eff} \cdot f_{NO_{eff}} \cdot Y_{RO_{2eff}}} \times \frac{NO2R}{VOCR}\right)^{-1} \tag{4}$$

The relative production of $RONO_2$ and $HNO_3$ is seen to be controlled by two factors, the first describing the chemistry of $RO_2$ radicals ($\alpha_{eff}$, $f_{NO_{eff}}$, $Y_{RO_{2eff}}$), and the second the ratio of NO2R to VOCR, which describes whether OH is more likely to react with a VOC or with $NO_2$. Because Eq. (4) concerns fractional loss of $NO_x$, the concentration of OH, which affects $RONO_2$ and $HNO_3$ production equally, does not appear in the result.

We show below that in the summertime continental boundary layer, the terms describing $RO_2$ radical chemistry vary significantly less than the NO2R/VOCR ratio, allowing the relative importance of $RONO_2$ and $HNO_3$ chemistry to be roughly estimated from only a single ratio.

## 3 Observed contributions of $HNO_3$ and $RONO_2$ chemistry to $NO_x$ loss

### 3.1 Daytime chemistry

Relative $RONO_2$ and $HNO_3$ production rates were calculated for 13 separate campaign deployments in the northern hemisphere over the past 20 years. Campaigns were selected that included measurements of $NO_x$, $HNO_3$, $O_3$, HCHO, a wide range of VOCs, and total organic nitrates ($\Sigma RONO_2$). Although they do not include measurements of $\Sigma RONO_2$, ITCT2k2 and CALNEX-P3 were also included to provide a pair of measurements of VOCs and $NO_x$ in the same geographic location separated in time. A list of all campaigns used in this study is given in Table 1. Where available, measurements of OH and $HO_2$ were used to directly calculate $RO_2$ formation and loss. When these radicals were not available, OH, and $HO_2$ radical concentrations were also calculated iteratively based on the total rate of $HO_x$ radical production by $O_3$ photolysis, HCHO photolysis, and alkene ozonolysis. When HONO was measured, HONO photolysis was also included as an OH source. In a small fraction of cases (3% of all data points), NO measurements were not available and NO concentrations were calculated based on the concentrations of $O_3$ and $NO_2$. Details of the radical modeling, including the equations used to calculate the production and loss of these radicals, are given in Appendix A.

Although these field campaigns do not constitute a random sample of the atmosphere, the combined dataset provides an excellent survey of atmospheric chemistry over a wide range of conditions. The combined dataset includes nearly 8000 data points for which fractional $NO_x$ loss can be calculated, spanning nearly 3 orders of magnitude in the ratio of NO2R to VOCR with no significant gaps (Fig. 2).

**Table 1.** Field campaigns used in this analysis

| Campaign name | Data Reference | Format | Year | Base of Operations | Date |
|---|---|---|---|---|---|
| ITCT2k2 | ITCT Science Team (2002) | Airborne | 2002 | Monterey, CA | 22 Apr – 19 May |
| INTEX-NA | INTEX-A Science Team (2006) | Airborne | 2004 | Palmdale, CA | 2 Jul |
| | | | | Mascoutah, IL | 7 Jul – 14 Jul |
| | | | | Portsmouth, NH | 16 Jul – 10 Aug |
| | | | | Mascoutah, IL | 12 Aug |
| INTEX-B | INTEX-B Science Team (2011) | Airborne | 2006 | Houston, TX | 4 Mar – 19 Mar |
| | | | | Honolulu, HI | 23 Apr – 28 Apr |
| | | | | Anchorage, AK | 1 May – 12 May |
| BEARPEX 2007 | BEARPEX 07 Science Team (2007) | Ground | 2007 | Georgetown, CA | 15 Aug –10 Oct |
| ARCTAS-B | ARCTAS-B Science Team (2011) | Airborne | 2008 | Palmdale, CA | 18 Jun – 24 Jun |
| | | | | Cold Lake, Alberta, CAN | 29 Jun – 8 Jul |
| | | | | Thule, Greenland | 8 Jul – 10 Jul |
| BEARPEX 2009 | BEARPEX 09 Science Team (2009) | Ground | 2009 | Georgetown, CA | 15 Jun – 31 Jul |
| CALNEX-P3 | CALNEX Science Team (2002a) | Airborne | 2010 | Ontario, CA | 1 May – 22 Jun |
| CALNEX-SJV | CALNEX Science Team (2002b) | Ground | 2010 | Bakersfield, CA | 15 May – 30 Jun |
| DC3 | DC3 Science Team (2013) | Airborne | 2012 | Salina, KS | 13 May – 30 Jun |
| SOAS | SOAS Science Team (2013) | Ground | 2013 | Centreville, AL | 1 Jun – 15 Jul |
| SEAC4RS | SEAC4RS Science Team (2014) | Airborne | 2013 | Houston, TX | 8 Aug – 23 Sep |
| FRAPPÉ | FRAPPÉ Science Team (2014) | Airborne | 2014 | Broomfield, CO | 16 Jul –16 Aug |
| KORUS-AQ | KORUS-AQ Science Team (2018) | Airborne | 2016 | Pyeongtaek, ROK | 1 May – 14 Jun |
| | | | | Palmdale, CA | 17 Jun – 18 Jun |

The fraction of total $NO_x$ loss occurring via $RONO_2$ chemistry from all 13 of these campaigns is shown in Fig. 3a for points within the continental summertime boundary layer. Despite spanning a large range of environments, all 13 campaigns are well described by a single function of the form $(1 + b \cdot (\frac{NO2R}{VOCR})^m)^{-1}$ (red line in Fig. 3a). This functional form corresponds to a linear relationship between $P(RONO_2)/P(HNO_3)$ and NO2R/VOCR on a log-log scale. If $m$ is fixed to 1, then this form

5  also corresponds to the expected behavior if the VOC mixture did not change between environments, and so all parameters other than NO2R/VOCR remained constant (gray line in Fig. 3a).

The calculated increase in fractional $NO_x$ loss via $RONO_2$ chemistry as NO2R/VOCR decreases is matched by an increase in the observed ratio of $\Sigma RONO_2$ to the sum of $\Sigma RONO_2$ and $HNO_3$ (Fig. 3b). However, the increase in fractional concentrations as NO2R/VOCR decreases is much less than the increase in fractional production. At low NO2R/VOCR ratios, the

10  dominant $RONO_2$ species are typically short lived and can undergo heterogeneous hydrolysis to produce $HNO_3$ (e.g., Browne

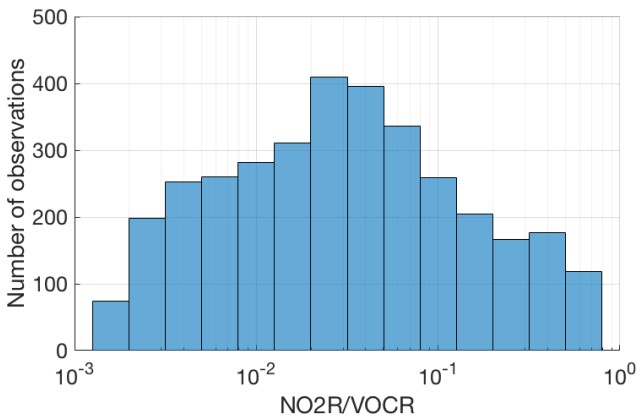

**Figure 2.** Number of points in each bin for which the fraction of $NO_x$ loss occurring via $RONO_2$ chemistry could be calculated.

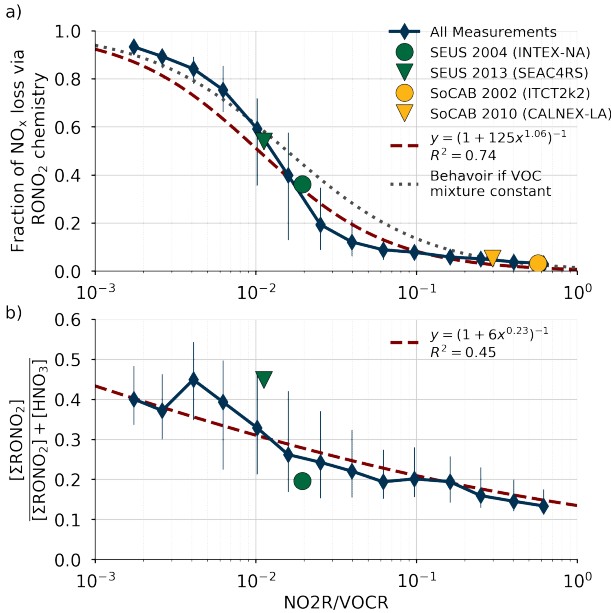

**Figure 3.** Comparison of the relative production rates of $RONO_2$ and $HNO_3$ as a function of NO2R/VOCR. Used data points are restricted to the continental summer daytime boundary layer (i.e., over land, less than 1.5 km above ground level, and average temperature $> 10\ °C$). The top panel shows the fraction of $NO_x$ loss attributable to $RONO_2$ chemistry, as well as a least-squares fit to the data and the expected behavior if $\alpha_{eff}$, $f_{NO_{eff}}$, $Y_{RO_{2\,eff}}$ were constant. The bottom panel shows the ratio of $\Sigma RONO_2$ to the sum of $HNO_3$ and $\Sigma RONO_2$. In each panel, the blue diamonds show the median in each bin and the vertical lines show the interquartile range.

et al., 2013). This indirect source of $HNO_3$ can be the greatest source of $HNO_3$ in forested environments, and leads to the relatively weak dependence of fractional concentration on NO2R/VOCR.

While the fraction of $NO_x$ loss occurring via $RONO_2$ chemistry can be well predicted from just the NO2R/VOCR ratio, the observations exhibit a sharper transition from $HNO_3$-dominated to $RONO_2$-dominated $NO_x$ loss than would be expected if the VOC mixture remained constant. This effect can be explained by variation in $Y_{RO_{2eff}}$, $\alpha_{eff}$, and $f_{NO_{eff}}$ as NO2R/VOCR changes. The behavior of these three parameters is shown in Fig. 4. As NO2R/VOCR decreases, $f_{NO_{eff}}$ consistently decreases from 0.8 to 0.2, due almost entirely to the decrease in $NO_x$ concentrations. In contrast, both $Y_{RO_{2eff}}$ and $\alpha_{eff}$ are larger in areas with low NO2R/VOCR ratios, due to changes in the VOC mixture between environments. In areas where NO2R/VOCR is high, many of the predominant VOCs, including CO, HCHO, and aromatics, either do not produce $RO_2$ radicals when oxidized by OH or produce $RO_2$ radicals that do not efficiently produce organic nitrates, leading to the relatively low values of $Y_{RO_{2eff}}$ and $\alpha_{eff}$. In areas with low NO2R/VOCR ratios, the VOC mixture is often dominated by biogenic alkenes such as isoprene and monoterpenes that efficiently produce organic nitrates, leading to higher values of both $Y_{RO_{2eff}}$ and $\alpha_{eff}$. However, although variation in these parameters can help explain some of the observed behavior of fractional $NO_x$ loss, the overall variation is much smaller than the variation of the NO2R/VOCR ratio. Each of the three parameters varies by a factor of 4 or less, while the NO2R/VOCR ratio varies by a factor of 1000.

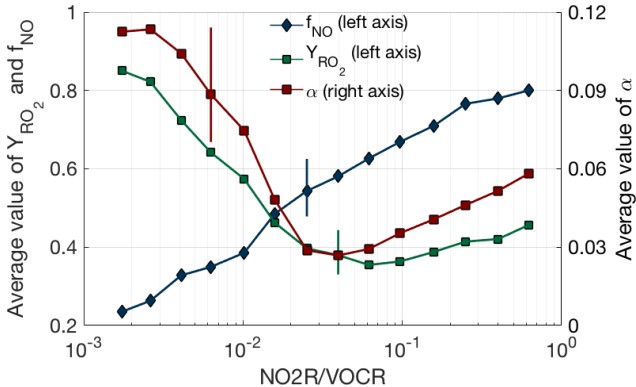

**Figure 4.** VOC oxidation parameters ($\alpha_{eff}$, $f_{NO_{eff}}$, $Y_{RO_{2eff}}$) as a function of NO2R/VOCR. Used data points are restricted to the continental summer daytime boundary layer (i.e., over land, less than 1.5 km above ground level, and average temperature > 10 °C). The line and solid shapes show the median in each bin, and the vertical lines show an example of the interquartile range for each binned parameter.

The conclusion that variation in VOC parameters is small compared to the variation in the NO2R/VOCR ratio does not hold outside of the summertime continental boundary layer. In the remote marine boundary layer or in the upper troposphere, $\alpha_{eff}$ is extremely low, as the dominant VOCs produce alkyl nitrates at yields of 0.01 or less (Mao et al., 2009; Perring et al., 2013). Under these conditions, $HNO_3$ dominates $NO_x$ loss even when NO2R/VOCR is less than $3 \times 10^{-2}$.

The trend calculated from the in situ observations matches that found in model simulations, that in areas with high ratios of NO2R to VOCR, $HNO_3$ is the dominant $NO_x$ sink, but as concentrations of $NO_x$ decrease and concentrations of VOCs increase, the opposite is true. The combined in situ observations show that the importance of $RONO_2$ chemistry to $NO_x$ loss is a non-linear function of the NO2R/VOCR ratio, leading to a sharp transition between the $HNO_3$-dominated and $RONO_2$-

dominated regimes. The sharp transition suggests there is a strong gradient in chemical $NO_x$ loss between urban and rural areas, especially in areas with significant biogenic VOC emissions. Furthermore, the sharp transition indicates that some regions may quickly shift from being $HNO_3$-dominated to $RONO_2$-dominated if NO2R/VOCR decreases.

## 3.2 Nighttime chemistry

While the primary focus of this analysis is on daytime chemistry, a conceptually similar transition may also occur at night. At night, OH concentrations are near zero, and the first step in $NO_x$ oxidation is the reaction of $NO_2$ with $O_3$ to produce $NO_3$. This radical can in turn react either with $NO_2$ to form $N_2O_5$ or with an alkene to form an organic nitrate (R8 – R9).

$$NO_3 + RH \rightarrow RONO_2 \tag{R8}$$

$$NO_3 + NO_2 \rightleftharpoons N_2O_5 \tag{R9}$$

Finally, $N_2O_5$ can either thermally decompose to reform $NO_3$ and $NO_2$ or it can hydrolyze on aerosol surfaces to produce $HNO_3$ (R9 – R10).

$$N_2O_5 \xrightarrow{k_{hyd}} 2\,HNO_3 \tag{R10}$$

Although the details of the nighttime chemical system are different, it shares some fundamental similarities with the daytime
system: $NO_x$ can be lost through the production of $RONO_2$ or of $HNO_3$, and a key step controlling the relative importance of these two sinks is whether an oxidant reacts with $NO_2$ or with a VOC. These similarities suggest that the relative importance of $RONO_2$ and $HNO_3$ as $NO_x$ sinks at night may be controlled by the relative reactivities of $NO_2$ and VOCs towards $NO_3$. In areas where $NO_3$ is more likely to react with $NO_2$, $HNO_3$ production is likely to dominate $NO_x$ loss, while the opposite is likely to be true in areas where $NO_3$ is more likely to react with a VOC.

However, quantitatively estimating the relative fraction of $NO_x$ lost through these different pathways is not practical with the combined dataset presented here. There have been relatively few measurements of the nocturnal atmosphere (only 4 of the 13 campaigns in Table 1 include nighttime measurements) and there remain significant uncertainties in the kinetics of nighttime $NO_x$ loss. In particular, the overall rate of $N_2O_5$ hydrolysis is controlled by the reactive uptake parameter $\gamma$ and the aerosol surface area, both of which can vary by multiple orders of magnitude (Brown et al., 2009; McDuffie et al., 2018). Variation in
the rate of $N_2O_5$ hydrolysis may therefore also play a major role in controlling the relative importance of $RONO_2$ and $HNO_3$ chemistry to $NO_x$ loss at night. While developing a more quantitative understanding of the trends in the chemical mechanisms of nocturnal $NO_x$ loss is an important area for future research, the conceptual similarity between the daytime and nighttime regimes suggests that conclusions based on daytime $NO_x$ chemistry may also be relevant to the nighttime.

## 4  Predicted trends over time

Using the trends in Fig. 3a to understand trends in $NO_x$ chemistry over time is only possible if the response to variation across space is equivalent to the response to variation across time. Two direct comparisons of fractional $NO_x$ loss in the same environment but at different times are found to fall along the same curve as the variation between campaigns in different locations (Fig. 3), indicating that such a substitution is valid in this analysis. The first case, INTEX-NA and SEAC4RS, sampled the southeast United States (SEUS) in 2004 and 2013; the second case, ITCT2k2 and CALNEX-P3, sampled the South Coast Air Basin (SoCAB) around Los Angeles in 2002 and 2010. Averages from these pairs of campaigns are shown in Fig. 3a and all four points fall along the same overall curve. For INTEX-NA and SEAC4RS, the shift in chemistry towards the $RONO_2$-dominated regime is accompanied by a dramatic shift in the ratio of $\Sigma RONO_2$ and $HNO_3$ concentrations, where $\Sigma RONO_2$ concentrations were only one quarter of $HNO_3$ in 2004 but were nearly equal to $HNO_3$ in 2013. $\Sigma RONO_2$ measurements are not available for ITCT2k2 or CALNEX-P3, preventing a similar comparison from being made for those campaigns.

Together, these cases indicate that the trend from Fig. 3a can be used to predict changes in fractional loss if the trend in NO2R/VOCR is known. Over the past decade, satellite measurements of $NO_2$ show a significant decrease in national $NO_2$ concentrations, reporting an average decrease of 4.5–7 % per year between 2005 and 2011 (Russell et al., 2012). No comparable satellite observations of VOCs exist, but studies in multiple locations have reported a decrease in primary anthropogenic VOC concentrations of 5.5–7.5 % per year over 2000-2010 (Geddes et al., 2009; Warneke et al., 2012; Pollack et al., 2013; Pusede et al., 2014). In contrast, biogenic VOC concentrations have been either constant or increasing over that same time period (Geddes et al., 2009; Hidy et al., 2014). Oxygenated VOCs show no major trend with time, although there are few long-term measurements of these species (Geddes et al., 2009; Pusede et al., 2014).

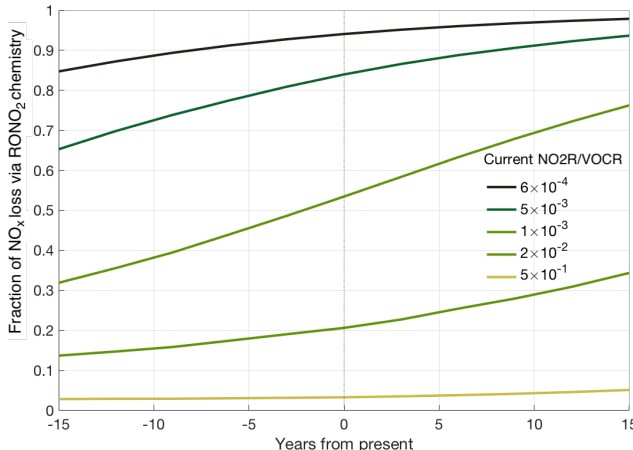

**Figure 5.** Predicted trends in fractional $NO_x$ loss over time, calculated from the estimated NO2R/VOCR ratio assuming a constant -6.5 % $yr^{-1}$ decrease in anthropogenic VOC concentrations, a 5.5 % $yr^{-1}$ decrease in $NO_x$ concentrations, and a 1.5 % $yr^{-1}$ increase in biogenic VOC concentrations.

These varied trends in $NO_x$, anthropogenic VOCs, and biogenic VOCs mean that NO2R/VOCR has not changed uniformly over the past decade. Past NO2R/VOCR ratios were calculated by assuming a 6.5 % $yr^{-1}$ decrease to anthropogenic VOC concentrations, a 5.5 % $yr^{-1}$ decrease to $NO_x$ concentrations, and a 1.5 % $yr^{-1}$ increase in biogenic VOC concentrations over the past 15 years. We also extrapolate these same trends to estimate NO2R/VOCR 15 years into the future. The calculated
NO2R/VOCR ratios are combined with the relationship from Fig. 3 to estimate fractional $NO_x$ loss at different times (Fig. 5). Based on these trends, $RONO_2$ chemistry is seen to have become a larger portion of total $NO_x$ loss over the past 15 years, although the change is not evenly distributed. The similar trends in $NO_x$ and anthropogenic VOCs cause there to have been little to no change in the regions with the highest NO2R/VOCR ratios (typically large cities). The largest changes are projected to occur in regions with moderate NO2R/VOCR ratios. In these regions, biogenic VOCs often account for a greater fraction
of the VOCR, leading to significant decreases in NO2R/VOCR over the past 15 years. In addition, the response of fractional $NO_x$ loss to changes in the NO2R/VOCR ratio is magnified in areas where both $RONO_2$ and $HNO_3$ chemistry contribute to $NO_x$ loss. In this transitional regime, if recent trends continue, the fraction of $NO_x$ loss occurring via $RONO_2$ chemistry could double in the next 15 years. Given the large number of data points sampled in this transition regime (Fig. 2), many regions of the United States are therefore likely to transition from a regime where $HNO_3$ dominates $NO_x$ loss to a mixed or
$RONO_2$-dominated regime.

## 5   Impacts of the transition from the $HNO_3$ to the $RONO_2$ regime

The growing importance of $RONO_2$ chemistry to $NO_x$ loss has several implications for air quality. Most directly, it means that understanding $NO_x$ chemistry in all but the most polluted megacities requires including the effects of $RONO_2$ chemistry. More theoretically, the transition from $HNO_3$- to $RONO_2$-dominated $NO_x$ loss affects how atmospheric chemistry will respond to
changes in emissions of $NO_x$ and VOCs. Because $RONO_2$ are produced in the same set of reactions that produce $O_3$, the fractional loss of $NO_x$ via $RONO_2$ chemistry is directly proportional to the ozone production efficiency (OPE), the ratio of ozone production to $NO_x$ loss (Eq. 5).

$$\text{OPE} = \frac{P(O_3)}{L(NO_x)} = \frac{2 \cdot \text{VOCR} \cdot Y_{RO_{2eff}} \cdot f_{NO_{eff}} \cdot (1 - \alpha_{eff})}{\text{NO2R} + \text{VOCR} \cdot Y_{RO_{2eff}} \cdot f_{NO_{eff}} \cdot \alpha_{eff}} \propto \frac{P(RONO_2)}{P(RONO_2) + P(HNO_3)} \tag{5}$$

Fundamentally, OPE represents the total amount of ozone produced for each molecule of $NO_x$ emitted. When considering
ozone pollution on regional scales, OPE is a more appropriate metric than instantaneous ozone production because it accounts for ozone production both locally and further afield.

Figure 6 uses the theoretic framework described in Romer et al. (2018) to investigate how ozone and $NO_x$ chemistry change as a function of NO2R/VOCR. As the NO2R/VOCR ratio decreases, OPE increases, reaching an inflection point exactly at the crossover point between the $HNO_3$-dominated and $RONO_2$-dominated regimes (Fig. 6a–b). For the polluted areas of the
country, where $HNO_3$ is currently the dominant $NO_x$ loss pathway, this means that, for example, interventions to improve air quality by reducing $NO_x$ emissions will be fighting uphill, because every incremental fractional decrease in $NO_x$ emissions will be associated with a growing incremental increase in OPE (Fig. 6c).

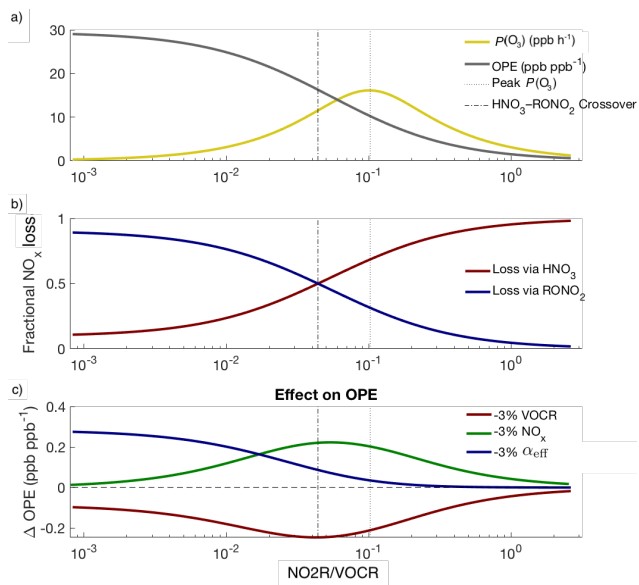

**Figure 6.** Theoretical picture of $NO_x$ and $O_3$ chemistry, calculated using variable $NO_x$ concentrations and fixed VOCR, $P(HO_x)$, and $\alpha_{eff}$. Panel a shows how $P(O_3)$ and OPE change as $NO_x$ changes; Panel b shows how the fractional $NO_x$ loss changes as NO2R/VOCR decreases; Panel c shows that changes to $NO_x$ and VOCR have their greatest effect on OPE not when $P(O_3)$ is at a maximum, but at the crossover point between the $RONO_2$-dominated and $HNO_3$-dominated regimes.

In addition, as $RONO_2$ chemistry becomes a more important part of the $NO_x$ budget, changes to $\alpha_{eff}$ have an increasing effect on OPE (Fig. 6c). Policy interventions that reduce VOCR but preferentially target high-$\alpha$ compounds (e.g., long-chain alkanes) could inadvertently increase ozone production or OPE (Farmer et al., 2011; Perring et al., 2013).

In addition to the large effects on aerosol yield that changes to $NO_x$ and VOC emissions have directly (e.g., Xu et al.,
2015a; Pusede et al., 2016), they also affect aerosols by changing the fate of $NO_x$. While both $HNO_3$ and $RONO_2$ can form aerosols (Stelson and Seinfeld, 1982; Pye et al., 2015), the properties of the resulting aerosols are likely to differ. Because $HNO_3$ is a strong acid, a shift towards $RONO_2$ chemistry is likely to increase aerosol pH. An increase in the role of $RONO_2$ chemistry will also cause more of the nitrate aerosol to be organic rather than inorganic, potentially affecting the viscosity and morphology of aerosols.

Further effects of changing $NO_x$ chemistry arise from the distinct fates of $RONO_2$ and $HNO_3$. Many $RONO_2$ compounds, especially those derived from isoprene, are remarkably reactive in the troposphere, with lifetimes of a few hours or less. A fraction of this $RONO_2$ loss returns $NO_x$ to the atmosphere, allowing $RONO_2$ production to effectively transport $NO_x$ downwind (Romer et al., 2016; Xiong et al., 2016). In contrast, $HNO_3$ is effectively chemically inert in the troposphere, with a chemical lifetime of 50 hours or more.

As a result of the differing chemical fates and lifetimes, transitioning from a $HNO_3$-dominated regime to a mixed or $RONO_2$-dominated regime has implications for the distribution of $NO_x$ on regional to continental scales. If a greater fraction

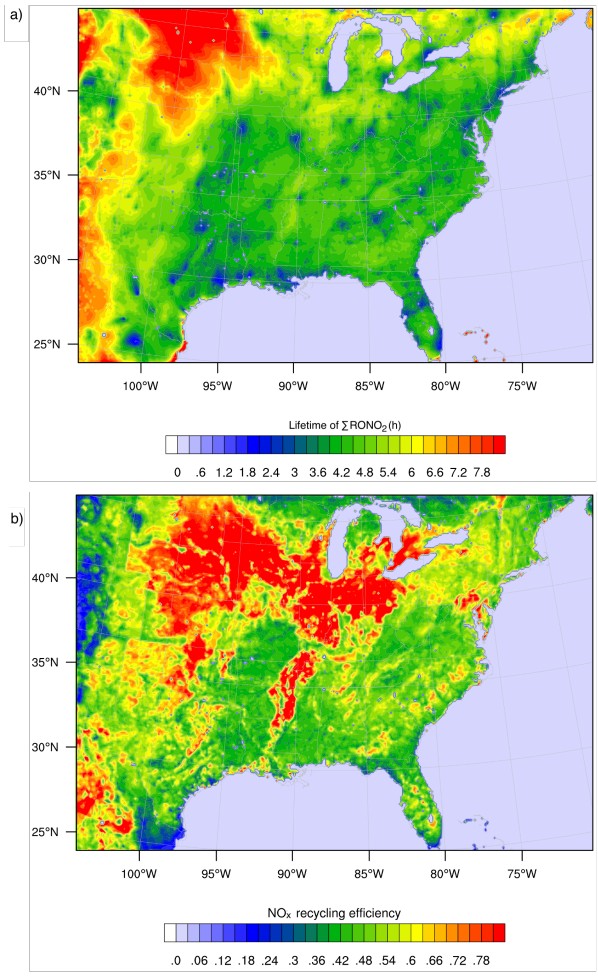

**Figure 7.** WRF-Chem simulation of $RONO_2$ chemistry over the southeast United States for summer 2013 as described in Zare et al. (2018). Panel a shows the overall lifetime of $\Sigma RONO_2$, defined as the concentration of $\Sigma RONO_2$ divided by their chemical loss rate for the daytime boundary layer. Panel b shows the average $NO_x$ recycling efficiency, defined as the local rate of $NO_x$ production from $RONO_2$ oxidation divided by the rate of $RONO_2$ production.

of $NO_x$ in polluted or moderately polluted regions is converted into $RONO_2$ compounds rather than into $HNO_3$, then more of the $NO_x$ may be re-released downwind, where it can participate in radical chemistry and ozone production. Simulations of $RONO_2$ chemistry using WRF-Chem and the RACM2_Berkeley2 mechanism (Zare et al., 2018) were used to investigate the $RONO_2$ lifetime and $NO_x$ recycling efficiency of $RONO_2$ across the southeast United States in summer 2013 (Fig. 7). Across
5   much of the region, $\Sigma RONO_2$ are calculated to have a lifetime of roughly 4 hours, and the release of $NO_x$ from $RONO_2$ oxidation was between 40 and 75 % of the instantaneous $RONO_2$ production rate. Combined, these findings demonstrate a significant role for $RONO_2$ chemistry in the transport of $NO_x$ between regions in the southeast United States. The effects of

organic nitrate chemistry on the distribution of $NO_x$ is likely to vary greatly across different regions of the United States and should be studied in further detail.

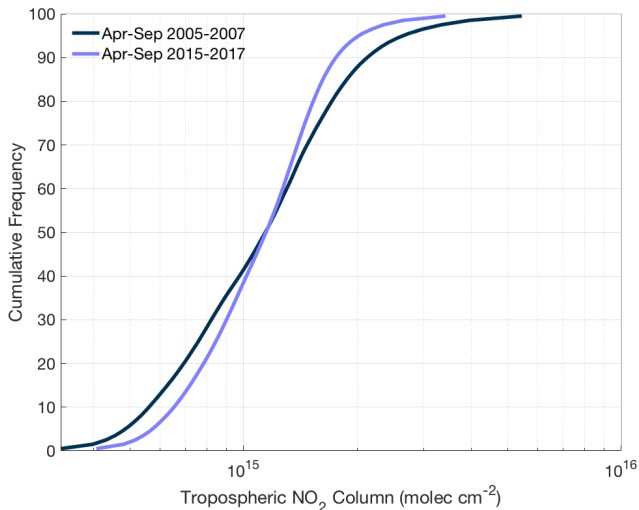

**Figure 8.** Cumulative frequency distribution of OMI tropospheric $NO_2$ columns over the continental United States using the BEHR v3.0A retrieval for summer (Apr–Sep) in 2005–2007 and 2015–2017.

Enhanced $NO_x$ transport between source and receptor regions is one aspect of a combined trend that is transforming the spatial distribution of $NO_x$. Over the past decade, $NO_x$ emission reductions have been concentrated in the most polluted environments. In these areas, motor vehicles and power plants, targets of emission control strategies, account for almost all of the $NO_x$ emissions. In less polluted regions, other sources of $NO_x$, including soil microbes (both in agricultural and non-agricultural regions), off-road vehicles, fires, and lightning, play a greater role in the $NO_x$ budget, reducing the effectiveness of typical combustion-related $NO_x$ emission controls. In addition, hemispheric background concentrations of $NO_x$ and $O_3$ have risen slightly over the past two decades (Cooper et al., 2012). The combination of all three of these trends suggests that the distribution of $NO_x$ across the United States is getting flatter over time. This trend matches satellite observations of $NO_2$ over the continental United States. Figure 8 shows the cumulative frequency distribution of summertime tropospheric $NO_2$ columns from 2005-2007 and 2015-2017 using the BErkeley High-Resolution (BEHR) v3.0A retrieval (Laughner et al., 2018) of slant-column measurements from OMI. Over this time, the highest percentiles of $NO_2$ concentrations have decreased and the lowest percentiles increased, leading to a significantly narrower distribution of $NO_2$ concentrations.

In summary, over past 15 years, decreases in anthropogenic $NO_x$ and VOC emissions have led to a significant shift in the mechanisms of daytime $NO_x$ loss. Many places where $HNO_3$ production dominated $NO_x$ loss are now mixed or have switched to a situation where the majority of $NO_x$ loss occurs through $RONO_2$ chemistry. If past trends continue, $RONO_2$ chemistry will grow to become an even more important fraction of $NO_x$ chemistry in coming decades. As a result of this combination of changing $NO_x$ chemistry, decreasing $NO_x$ emissions, and increasing background concentrations, air pollution in the United

States may transform from a highly local issue to a more extended regional one. Efforts to control air pollution focused only on local sources are less likely to be effective; future improvements in air quality and attaining the most recent National Ambient Air Quality Standards are likely to require coordinated efforts on regional scales to broadly reduce $NO_x$ emissions.

*Data availability.* Data from ARCTAS, DC3, FRAPPÉ, INTEX-NA, INTEX-B, KORUS-AQ and SEAC4RS are available from https://www-air.larc.nasa.gov/missions.htm. Data from ITCT2k2, CALNEX, SOAS, UBWOS, and WINTER are available from https://www.esrl.noaa.gov/csd/field.html. The BEHR retrieval of OMI $NO_2$ columns is available at http://behr.cchem.berkeley.edu/.

## Appendix A: Calculation of the $RONO_2$ production rate

### A1 Steady-state calculation of unmeasured radicals

The formation rates of $RONO_2$, $HNO_3$, and $O_3$ depend either directly or indirectly on the concentration of OH, $HO_2$, $RO_2$, NO, and $NO_2$. Speciated $RO_2$ radicals are not currently observable in the atmosphere, and thus all $RO_2$ concentrations were calculated assuming they were in steady-state, with their production and loss rates equal.

There were additional periods in which some combination of OH, $HO_2$, and NO were also not measured, and these radicals were also assumed to be in steady state. Concentrations of VOCs, $NO_2$, and $O_3$ were always taken from measurements. In order to calculate the steady-state concentrations of unmeasured radicals, reaction rate constants and $RO_2$ yields for the different VOCs were taken from the MCM v3.3.1 (Jenkin et al., 2015). Concentrations of all unmeasured species were calculated iteratively until all the concentrations converged. Equations (A1–A8) were used to calculate the steady-state concentration of unmeasured radicals. In Eq. (A8), the symbol XR is used to represent the OH reactivity of species such as $SO_2$ and $O_3$ that are not included in either VOCR or NO2R. Although it is not often categorized as a VOC, CO is included as a contributor to VOCR. The reaction rate constant for $NO_2$ with OH was taken from Mollner et al. (2010), with temperature- and pressure-dependencies from Henderson et al. (2012).

$$P(\mathrm{RO_2}) = [\mathrm{OH}] \cdot \mathbf{VOCR} \cdot Y_{\mathrm{RO_2}} \tag{A1}$$

$$L(\mathrm{RO_2}) = k_{\mathrm{RO_2+NO}}[\mathrm{RO_2}][\mathrm{NO}] + k_{\mathrm{RO_2+HO_2}}[\mathrm{RO_2}][\mathrm{HO_2}] + 2k_{\mathrm{RO_2+RO_2}}[\mathrm{RO_2}][\mathrm{RO_2}] + k_{\mathrm{isom}}[\mathrm{RO_2}] \tag{A2}$$

$$P(\mathrm{HO_2}) = k_{\mathrm{RO_2+NO}}[\mathrm{RO_2}][\mathrm{NO}](1-\alpha) + [\mathrm{OH}] \cdot \mathbf{VOCR} \cdot Y_{\mathrm{HO_2}} + 2j_{\mathrm{HCHO}}[\mathrm{HCHO}] \tag{A3}$$

$$L(\mathrm{HO_2}) = k_{\mathrm{HO_2+NO}}[\mathrm{HO_2}][\mathrm{NO}] + 2k_{\mathrm{HO_2+HO_2}}[\mathrm{HO_2}][\mathrm{HO_2}] + k_{\mathrm{HO_2+RO_2}}[\mathrm{HO_2}][\mathrm{RO_2}] \tag{A4}$$

$$P(\mathrm{NO}) = j_{\mathrm{NO_2}}[\mathrm{NO_2}] \tag{A5}$$

$$L(\mathrm{NO}) = k_{\mathrm{O_3+NO}}[\mathrm{O_3}][\mathrm{NO}] + k_{\mathrm{RO_2+NO}}[\mathrm{RO_2}][\mathrm{NO}] + k_{\mathrm{HO_2+NO}}[\mathrm{HO_2}][\mathrm{NO}] \tag{A6}$$

$$P(\mathrm{OH}) = \frac{2j_{\mathrm{O_3 \to O^1D}}[\mathrm{O_3}] \cdot k_{\mathrm{O^1D+H_2O}}[\mathrm{H_2O}]}{k_{\mathrm{O^1D+H_2O}}[\mathrm{H_2O}] + k_{\mathrm{O^1D}+M}[M]} + j_{\mathrm{HONO}}[\mathrm{HONO}] + k_{\mathrm{HO_2+NO}}[\mathrm{HO_2}][\mathrm{NO}] + k_{\mathrm{O_3+RH}}[\mathrm{O_3}][\mathrm{RH}]Y_{\mathrm{OH}} \tag{A7}$$

$$L(\mathrm{OH}) = (\mathbf{VOCR} + \mathbf{NO2R} + \mathbf{XR})[\mathrm{OH}] \tag{A8}$$

In order to test the accuracy of the modeling, we used periods when $HO_2$, OH, and NO were all measured and calculated how the production ratio $P(RONO_2)/P(HNO_3)$ changed if modeled radical concentrations were substituted for the measured values. These results are shown in Fig. A1. Even in the worst-case scenario (modeled concentrations used for all radicals), the slope is close to one (Fig. A1a), indicating that the use of modeled radicals does not significantly affect our results. Furthermore, Fig. A1b–d show that the use of modeled OH or $HO_2$ concentrations alone does not lead to noticeable changes in $P(RONO_2)/P(HNO_3)$. Use of modeled NO concentrations can cause small but noticeable changes in $P(RONO_2)/P(HNO_3)$, but modeled NO concentrations are used in less than 3% of all data points used in this analysis (238 out of 7988 data points).

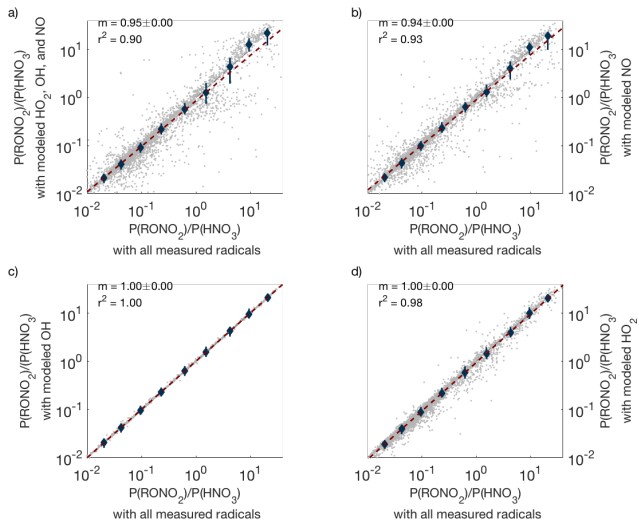

**Figure A1.** Comparison of $P(RONO_2)/P(HNO_3)$ when measured concentrations of all possible radicals are used ($x$-axis) versus when measured concentrations are replaced by modeled concentrations ($y$-axis). Panel a shows the result when modeled concentrations of OH, $HO_2$, and NO are all used simultaneously; Panels b–d show the effect of replacing measured with modeled values one species at a time.

## A2   Determination of $\alpha$

Accurately calculating the $RONO_2$ production rate requires accurate knowledge of $\alpha_i$ for all VOCs. If values of $\alpha$ had been reported for a specific compound from laboratory measurements, the most recent value was applied (Perring et al., 2013; Teng et al., 2015; Rindelaub et al., 2015; Praske et al., 2015; Wennberg et al., 2018). In cases where no reliable laboratory measurements are available, the parameterization for $\alpha$ from Wennberg et al. (2018) was used. In all cases, the temperature- and pressure-dependencies described in Wennberg et al. (2018) were used to scale the laboratory measurements of $\alpha$ to the conditions of the atmosphere.

*Author contributions.* PSRP and RCC designed the experiment, PSRP performed the analysis of field campaign data and wrote the paper with contributions from all authors, AZ designed, ran, and analyzed the modeling simulations, and RCC supervised the project.

*Competing interests.* The authors declare that they have no conflict of interest.

*Acknowledgements.* This study was supported by NOAA grant NA18OAR4310117 in the Atmospheric Chemistry, Carbon Cycle, and Climate program of the NOAA Climate Program Office and by NSF grant AGS-1352972. The authors thank Joshua Laughner for assistance with the OMI BEHR retrieval.

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
