# Peer review of "The changing role of organic nitrates in the removal and transport of $\mathbf{NO}_{\mathbf{x}}$"

_Atmospheric Chemistry and Physics, 2019_

## Referee Comment (RC1) · Anonymous Referee #1 · 6 Jun 2019

This work investigates the changing fate of atmospheric nitrogen oxides (NOx), with a focus on the increasing importance of the role of organic nitrates. The authors develop a framework by which the fate of NOx can be interpreted using the ratio of contributions to the hydroxyl radical loss rate from nitrogen dioxide and volatile organic hydrocarbons (VOC). The method is then demonstrated using observations from 13 separate field studies, predominantly in the US, dating back over the previous 20 years. The authors then use the framework to investigate the changing role of organic nitrates in the continental US, and the implications for air quality policy. This results in some interesting and important conclusions about the future of NOx chemistry in the US. This is an interesting approach to the study of the non-linear sensitivities of atmospheric NOx chemistry and provides another way by which to explore the importance and im-

pacts of high and low NOx chemical regimes. The approach will also likely prove to be a useful tool for assessing model organic nitrates and responses to changing emissions. The paper is well written and represents a valuable contribution to the literature. I recommend publication after the following comments have been addressed.

Comments:

1) Figure 2 is a very powerful and novel presentation of historical field observations, and warrants more discussion. The authors use of a fit function of the same form as eqn. 4 is convenient for the purposes of this study, but it would be useful to the reader to know the form of a best fit function to the observational data. The delay seen in the shift from inorganic to organic nitrate dominated NOx loss, as NO2R/VOCR decreases, compared with the best fit line and that predicted by the authors using a fixed VOC speciation (gray line in Fig. 2a) is interesting. This discrepancy occurs at the transition between the high and low NOx regimes identified, i.e. inorganic vs organic nitrate production dominated, and is thus the most important region to understand. It is suggested by the authors that this discrepancy is likely due a change in the effective branding ratio fo organic nitrate production (page 5 line 28), however this needs to be supported. Could it not also be due to a change in the reactivity of the VOC mixture, and therefore VOCR, or changes in fNO? The authors should discuss further the observational trends shown in Fig. 2, and the possible explanations for the discrepancy with the simple fit shown in the red line and with the calculated gray line in Fig. 2a. The authors should also comment on the implications of this discrepancy for the conclusions of the paper.

2) It would be very useful to the reader to understand how the high and low NOx regimes identified in this work differ from the traditional high to low NOx definition used for describing ozone production regimes (e.g. Jaeglé et al. 1999; Thornton et al. 2002). An important parameter in the traditional conceptual model of high and low NOx chemistry is if the dominant peroxy radical reaction partner is NO or another peroxy radical (i.e. the author defined fNO). In Appendix A the authors have already

calculated radical production and loss, and could use this to estimate fNO for each set of observations. It would be helpful for the reader if these values were shown, possibly on Fig. 2a, so the two definitions of high and low NOx chemistries could be compared.

3) Although the assumptions made in the calculations in Appendix A are commonly used, it is also frequently the case that incomplete measurement suites result in discrepancies with observations. As the authors have radical measurements for some of the field data, comparisons with calculated values for these studies would provide a gauge of the uncertainty in the calculations. The authors should also provide an estimate to which this uncertainty impacts the conclusions of the paper.

4) This work focuses on the daytime production of organic vs inorganic nitrates. Perring et al. (2013), however, estimate that the nocturnal production of organic nitrates, via nitrate radical reactions with alkenes and phenols, could account for as much as 50% of regional production. As NOx and VOC emissions change is the importance of this nocturnal pathway to organic nitrate production also likely to change? A recent analysis by Edwards et al. (2017) presented a similar relationship to that shown here, relating nocturnal organic vs inorganic nitrate formation to the ratio of NOx to VOC, and also predicted a transition from inorganic to organic nitrate dominance. The authors should discuss the role of nocturnal organic nitrate production and how this is accounted for and impacts on the conclusions of this work.

References: Jaeglé et al. J. Geophys. Res., 26, 29, 3081-3084, 1999.; Thornton et al. J. Geophys. Res., 107, 12, 4146, 2002.; Perring et al., Chem. Rev., 113, 8, 5848-5870, 2013.; Edwards et al. Nat. Geosci., 10, 7, 490-495, 2017.

---

## Referee Comment (RC2) · Anonymous Referee #2 · 11 Jun 2019

This paper evaluated the relative importance of two NOx removal pathways, forming HNO3 and RONO2. The fraction of NOx loss via RONO2 chemistry was approximated the contribution of VOC to the total OH reactivity with additional consideration of RONO2 yield. The comparison of such simple calculation and 13 field campaigns results show consistent trend, which give confidence to conduct long-term prediction. The impact on the ozone production is discussed based on the growing importance of RONO2 chemistry to NOx loss. Finally, the spatial distribution of NOx lifetime is evaluated using WRF-Chem model for the 2013 summer United States. This paper presents an interesting result on the fate of NOx using a simplified but insightful approach. The paper is well written and the method to evaluate the fate the NOx is helpful to diagnose the non-linearity of the atmospheric chemistry. I recommend publication after

the following comments are addressed. Comments: (1) The uncertainty in the radical budget calculation. In the appendix, the production of OH, HO2 and RO2 are not complete. The HONO photolysis, alkene ozonolysis are missing. How much does the result rely on this budget analysis? Some results showed that HONO photolysis, as an OH source, is more important than ozone photolysis in polluted environments (Mao et al., 2010;Tan et al., 2019). Ozonolysis of alkenes, isoprene, and monoterpenes could be important radical source in forest areas (Griffith et al., 2013). A discussion on this missing radical source and its impact would be helpful. Also, I assume these radical concentrations are used to calculate the P(HNO3) and P(RONO2) separately for each 13 campaigns but it's not clear in the context. The authors should make this point clearer. (2) The detail description of alpha. The organic nitrate yield is determined for different VOCs and explained in the appendix. However, I would suggest adding a table to describe the range of organic nitrate yield. As I found in Figure 2, the least-squares fit is y=(1+125x^1.06)^{-1} and 125 should be equal to 1/(alpha*fNO*YRO2), the alpha would be 0.008 if fNO and YRO2 are unity. In this case, a comparison with least-squares fit to Eq. 4 and discussion on the parameters would be helpful to the reader to understand what the meaning of such fit function is. (3) Some argumentation are too general or without explanation. Page 4 Line 23, please define low-NOx Page 9 Line 1-2, according to Fig. 3, this statement relies on an assumption that many regions are located in the transition regime (e.g. NO2R/VOCR ranges from 2e-2 to 1e-3). The authors should provide relevant information to support this argument. Page 9 Line12, please define comprehensive metric. Technical corrections: Page 5 Line 1, 'NO2R' 2 should be subscripted. Page 5 Line 2, please provide the reaction rate constant for OH+NO2 reaction and the literature. Page 9 Line 10, 'P(O3)' 3 should be subscripted. Page 13 Line 14, (A3) 2jHCHO*[HCHO] Page 13 Line 19, (A8) L(OH) should be (VOCR+NO2R)*[OH]

References: Griffith, S. M., Hansen, R. F., Dusanter, S., Stevens, P. S., Alaghmand, M., Bertman, S. B., Carroll, M. A., Erickson, M., Galloway, M., Grossberg, N., Hottle, J., Hou, J., Jobson, B. T., Kammrath, A., Keutsch, F. N., Lefer, B. L., Mielke,

L. H., O'Brien, A., Shepson, P. B., Thurlow, M., Wallace, W., Zhang, N., and Zhou, X. L.: OH and HO2 radical chemistry during PROPHET 2008 and CABINEX 2009-Part 1: Measurements and model comparison, Atmos. Chem. Phys., 13, 5403-5423, https://doi.org/10.5194/acp-13-5403-2013, 2013. Mao, J., Ren, X., Chen, S., Brune, W. H., Chen, Z., Martinez, M., Harder, H., Lefer, B., Rappenglueck, B., Flynn, J., and Leuchner, M.: Atmospheric oxidation capacity in the summer of Houston 2006: Comparison with summer measurements in other metropolitan studies, Atmos. Environ., 44, 4107-4115, 10.1016/j.atmosenv.2009.01.013, 2010. Tan, Z., Lu, K., Jiang, M., Su, R., Wang, H., Lou, S., Fu, Q., Zhai, C., Tan, Q., Yue, D., Chen, D., Wang, Z., Xie, S., Zeng, L., and Zhang, Y.: Daytime atmospheric oxidation capacity in four Chinese megacities during the photochemically polluted season: a case study based on box model simulation, Atmos. Chem. Phys., 19, 3493-3513, 10.5194/acp-19-3493-2019, 2019.

---

## Referee Comment (RC3) · Anonymous Referee #3 · 19 Jun 2019

This paper makes use of the available data from field campaign both on the ground and on aircrafts in the USA to explore how the general decrease of anthropogenic emissions (both NOx and VOCs) is affecting the ozone production by increasing the importance of RONO2 chemistry compared to the NOx loss. The study is well presented and makes a clever use of past available data obtaining what looks like a relatively robust tool to make future predictions.

This referee agrees with what already suggested by the other reviewers. In particular the point regarding the calculation of the OH, HO2 and RO2 and radicals should be better discussed. As pointed out a better analysis of the uncertainties for the calculation should be done together with the inclusion, if possible, of HONO photolysis and ozonolysis of unsaturated compounds. The calculation, as is at the moment, is

very simplified (for example, why not including reaction with CO when considering the losses of OH radicals?) and it can well be that it is good enough for this study but a sensitivity check by adding additional sources would help understand their impact. In addition, the comparison, where possible, with the available radical measurements would also help understanding the reliability of the simple calculation used.

---

## Author Comment (AC1) · 9 Sep 2019

**Response to Reviewer 1**

We thank the reviewer for their helpful comments.

This work investigates the changing fate of atmospheric nitrogen oxides (NOx), with a focus on the increasing importance of the role of organic nitrates. The authors develop a framework by which the fate of NOx can be interpreted using the ratio of contributions to the hydroxyl radical loss rate from nitrogen dioxide and volatile organic hydrocarbons (VOC). The method is then demonstrated using observations from 13 separate field studies, predominantly in the US, dating back over the previous 20 years. The authors then use the framework to investigate the changing role of organic nitrates in the continental US, and the implications for air quality policy. This results in some interesting and important conclusions about the future of NOx chemistry in the US. This is an interesting approach to the study of the non-linear sensitivities of atmospheric NOx chemistry and provides another way by which to explore the importance and impacts of high and low NOx chemical regimes. The approach will also likely prove to be a useful tool for assessing model organic nitrates and responses to changing emissions. The paper is well written and represents a valuable contribution to the literature. I recommend publication after the following comments have been addressed.

**Comments:**

1) Figure 2 is a very powerful and novel presentation of historical field observations, and warrants more discussion. The authors use of a fit function of the same form as eqn. 4 is convenient for the purposes of this study, but it would be useful to the reader to know the form of a best fit function to the observational data.

In the course of our analysis we have tested several different methods of fitting the data shown in Fig. 2. Multiple forms, including the one presented in Fig. 2, had similar ability to explain the observations (as measured by  $r^2$  or by the standard deviation of the residuals), with no fit clearly superior to the rest. The form presented in Fig. 2 was chosen to be the focus of further analysis both because it matches our theoretical understanding of the process and because it corresponds to a linear relationship between NO2R/VOCR and P(RONO2)/P(HNO3) on a loglog scale. We have added text describing these points:

**Page 6:** "Despite spanning a large range of environments, all 13 campaigns are well described by a single function of the form  $(1 + b \cdot (NO2R/VOCR)^m)^{-1}$  (red line in Fig. 3a). This functional form corresponds to a linear relationship between P(RONO2)/P(HNO3) and NO2R/VOCR on a log-log scale. If m is fixed to 1, then this form also corresponds to the expected behavior if the VOC mixture did not change between environments, and so all parameters other than NO2R/VOCR remained constant (gray line in Fig. 3a)."

The delay seen in the shift from inorganic to organic nitrate dominated NOx loss, as NO2R/VOCR decreases, compared with the best fit line and that predicted by the authors using a fixed VOC speciation (gray line in Fig. 2a) is interesting. This discrepancy occurs at the transition between the high and low NOx regimes identified, i.e. inorganic vs organic nitrate production dominated, and is thus the most important region to understand. It is suggested by the authors that this discrepancy is likely due a change in the effective branding ratio of organic nitrate production (page 5 line 28), however this needs to be supported. Could it not also be due to a change in the reactivity of the VOC mixture, and therefore VOCR, or changes in fNO? The authors should discuss further the observational trends shown in Fig. 2, and the possible explanations for the discrepancy with the simple fit shown in the red line and with the calculated gray line in Fig. 2a. The authors should also comment on the implications of this discrepancy for the conclusions of the paper.

While changes in the VOC mixture do have an importance effect on VOCR, we have already taken this into account in our analysis through the use of NO2R/VOCR as our x-variable. Therefore, changes in the VOC mixture cannot explain the deviations between the observations and the best-fit line. Changes in any of fNO, YRO2, or alpha are a more likely explanation for the discrepancies. We have revised our discussion to discuss this point and included an additional figure showing how all three of these parameters vary with NO2R/VOCR:

**Page 8:** "While the fraction of NOx loss occurring via RONO2 chemistry

can be well predicted from just the NO2R/VOCR ratio, the observations exhibit a sharper transition from HNO3-dominated to RONO2-dominated  $NO_x$  loss than would be expected if the VOC mixture remained constant. This effect can be explained by variation in  $Y_{RO_{2eff}}$ ,  $\alpha_{eff}$ , and  $f_{NO_{eff}}$  as NO2R/VOCR changes. The behavior of these three parameters is shown in Fig. 4. As NO2R/VOCR decreases, fNOeff consistently decreases from 0.8 to 0.2, due almost entirely to the decrease in NOx concentrations. In contrast, both  $Y_{RO_{2}}$  and  $\alpha_{eff}$  are larger in areas with low NO2R/VOCR ratios, due to changes in the VOC mixture between environments. In areas where NO2R/VOCR is high, many of the predominant VOCs, including CO, HCHO, and aromatics, either do not produce RO2 radicals when oxidized by OH or produce RO2 radicals that do not efficiently produce organic nitrates, leading to the relatively low values of both these parameters. In areas with low NO2R/VOCR ratios, the VOC mixture is often dominated by biogenic alkenes such as isoprene and monoterpenes that efficiently produce organic nitrates, leading to higher values of both  $Y_{RO_{2eff}}$  and  $\alpha_{eff}$ . However, although variation in these parameters can help explain some of the observed behavior of fractional  $NO_x$  loss, the overall variation is much smaller than the variation of the NO2R/VOCR ratio. Each of the three parameters varies by a factor of 4 or less, while the NO2R/VOCR ratio varies by a factor of 1000."

**Figure 4:** "VOC oxidation parameters ( $\alpha_{eff}$ ,  $f_{NO_{eff}}$ ,  $Y_{RO_{2eff}}$ ) as a function of NO2R/VOCR. Used data points are restricted to the continental summer daytime boundary layer (i.e., over land, less than 1.5 km above ground level, and average temperature > 10 °C). The line and solid shapes show the median in each bin, and the vertical lines show an example of the interquartile range for each binned parameter."

2) It would be very useful to the reader to understand how the high and low NOx regimes identified in this work differ from the traditional high to low NOx definition used for describing ozone production regimes (e.g. Jaeglé et al. 1999; Thornton et al. 2002). An important parameter in the traditional conceptual model of high and low NOx chemistry is if the dominant peroxy radical reaction partner is NO or another peroxy radical (i.e. the author defined fNO). In Appendix A the authors have already calculated radical production and loss, and could use this to estimate fNO for each set of observations. It would be helpful for the reader if these values were shown, possibly on Fig. 2a, so the two definitions of high and low NOx chemistries could be compared. The regimes identified in this paper, of  $HNO_3$ -dominated and  $RONO_2$ dominated  $NO_x$  loss, do not exactly correspond either to the  $NO_x$ saturated/ $NO_x$ -limited regimes of ozone production or to the high-  $NO_x$ /low- $NO_x$  regimes of peroxy radical chemistry. To avoid confusion, we have removed the terms "high-  $NO_x$ " and "low-  $NO_x$  " from our manuscript. A graph of fNO is also included in the new Figure 4 in our manuscript.

3) Although the assumptions made in the calculations in Appendix A are commonly used, it is also frequently the case that incomplete measurement suites result in discrepancies with observations. As the authors have radical measurements for some of the field data, comparisons with calculated values for these studies would provide a gauge of the uncertainty in the calculations. The authors should also provide an estimate to which this uncertainty impacts the conclusions of the paper.

All three of the reviewers highlighted the need for additional discussion of the effects of the radical modeling on our results. We have therefore added a section to Appendix A discussing the effects of the steady-state radical modeling on our results:

**Page 16:** "In order to test the accuracy of the modeling, we used periods when HO2, OH, and NO were all measured and calculated how the production ratio  $P(\text{RONO}_2)/P(\text{HNO}_3)$  changed if modeled radical concentrations were used instead. These results are shown in Fig. A1. Even in the worst-case scenario (modeled concentrations used for all radicals), the slope is close to one (Fig. A1a), indicating that the use of modeled radicals does not significantly affect our results. Furthermore, Fig A1b–d show that the use of modeled OH or HO2 concentrations alone does not lead to noticeable changes in  $P(\text{RONO}_2)/P(\text{HNO}_3)$ . Use of modeled NO concentrations can cause small but noticeable changes in  $P(\text{RONO}_2)/P(\text{HNO}_3)$ , but modeled NO concentrations are used in less than 3% of all data points used in this analysis (238 out of 7988 data points)."

**Figure A1**: "Comparison of  $P(\text{RONO}_2)/P(\text{HNO}_3)$  when measured concentrations of all possible radicals are used (*x*-axis) versus when measured concentrations are replaced by modeled concentrations (*y*-axis). Panel a shows the result when modeled concentrations of OH, HO2, and NO are all used simultaneously; Panels b–d show the effect of replacing measured with modeled values one species at a time."

4) This work focuses on the daytime production of organic vs inorganic nitrates. Perring et al. (2013), however, estimate that the nocturnal production of organic nitrates, via nitrate radical reactions with alkenes and phenols, could account for as much as 50% of regional production. As NOx and VOC emissions change is the importance of this nocturnal pathway to organic nitrate production also likely to change? A recent analysis by Edwards et al. (2017) presented a similar relationship to that shown here, relating nocturnal organic vs inorganic nitrate formation to the ratio of NOx to VOC, and also predicted a transition from inorganic to organic nitrate production and how this is accounted for and impacts on the conclusions of this work.

Nighttime oxidation is likely to play an important in the overall rate of  $NO_x$  loss. We have performed qualitative analyses that suggest that as the ratio of NO2R/VOCR decreases, RONO2 production is likely to become a greater fraction of nocturnal  $NO_x$  loss. However, quantitatively testing these results is extremely difficult for two reasons. First, observations of nighttime chemistry, particularly in the residual layer, are extremely limited, making it hard to examine trends over time. Second, the kinetics of nighttime  $NO_x$  chemistry are generally more variable and less well understood than daytime  $NO_x$  chemistry. In particular, the value of the reactive uptake coefficient for  $N_2O_5$  on aerosols can vary by several orders of magnitude and is not well predicted by current models, making it difficult to predict how  $NO_x$  will be lost in different environments. A more full examination of trends in the nighttime mechanics of  $NO_x$  loss is an important topic for further research, but one that we feel is better suited to its own analyses than to be included in this paper.

We have added a section to this paper describing nocturnal  $NO_x$  chemistry, the similarities between the daytime and nighttime oxidation mechanisms, and the difficulties of extending the analysis from Fig. 2 into the night:

**Page 9:** "While the primary focus of this analysis is on daytime chemistry, a conceptually similar transition may also occur at night. At night, OH concentrations are near zero, and the first step in  $NO_x$  oxidation is the reaction of  $NO_2$  with  $O_3$  to produce  $NO_3$ . This radical can in turn react either with  $NO_2$  to form  $N_2O_5$  or with an alkene to form an organic nitrate (R8 – R9).

 $NO_3 + RH \rightarrow RONO_2 (R8)$

 $NO_3+NO_2 \leftrightarrow N_2O_5(R9)$

Finally,  $N_2O_5$  can either thermally decompose to reform  $NO_3$  and  $NO_2$  or it can hydrolyze on aerosol surfaces to produce  $HNO_3$  (R9 – R10).

 $N_2O_5 \xrightarrow{k_{hyd}} 2HNO_3 (R10)$

Although the details of the nighttime chemical system are different, it shares some fundamental similarities with the daytime system:  $NO_x$  can be lost through the production of RONO2 or of HNO3, and a key step controlling the relative importance of these two sinks is whether an oxidant reacts with  $NO_2$  or with a VOC. These similarities suggest that the relative importance of RONO2 and HNO3 as  $NO_x$  sinks at night may also be controlled by the relative reactivities of  $NO_2$  and VOCs towards  $NO_3$ . In areas where  $NO_3$  is more likely to react with  $NO_2$ , HNO3 production is likely to dominate  $NO_x$ loss, while the opposite is likely to be true in areas where  $NO_3$  is more likely to react with a VOC.

However, quantitatively estimating the relative fraction of  $NO_x$  loss through these different pathways is not practical with the combined dataset presented here. There have been relatively few measurement of the nocturnal atmosphere (only 4 of the 13 campaigns in Table 1 include nighttime measurements) and there remain significant uncertainties in the kinetics of nighttime  $NO_x$  loss. In particular, the overall rate of  $N_2O_5$  hydrolysis is controlled by the reactive uptake parameter  $\gamma$  and the aerosol surface area, both of which can vary by multiple orders of magnitude (McDuffie et al., 2018; Brown et al., 2009). Variation in the rate of  $N_2O_5$  hydrolysis may therefore also play a major role in controlling the relative importance of RONO2 and HNO3 production to NOx loss at night. While developing a more quantitative understanding of the trends in the chemical mechanisms of nocturnal NOx loss is an important area for future research, the conceptual similarity between the daytime and nighttime regimes suggests that the conclusions drawn here based on the daytime chemistry may also be relevant to the nighttime. "

**Response to Reviewer 2**

We thank the reviewer for their helpful comments.

This paper evaluated the relative importance of two NOx removal pathways, forming HNO3 and RONO2. The fraction of NOx loss via RONO2 chemistry was approximated the contribution of VOC to the total OH reactivity with additional consideration of RONO2 yield. The comparison of such simple calculation and 13 field campaigns results show consistent trend, which give confidence to conduct long-term prediction. The impact on the ozone production is discussed based on the growing importance of RONO2 chemistry to NOx loss. Finally, the spatial distribution of NOx lifetime is evaluated using WRF-Chem model for the 2013 summer United States. This paper presents an interesting result on the fate of NOx using a simplified but insightful approach. The paper is well written and the method to evaluate the fate the NOx is helpful to diagnose the non-linearity of the atmospheric chemistry. I recommend publication after the following comments are addressed. Comments:

(1) The uncertainty in the radical budget calculation. In the appendix, the production of OH, HO2 and RO2 are not complete. The HONO photolysis, alkene ozonolysis are missing. How much does the result rely on this budget analysis? Some results showed that HONO photolysis, as an OH source, is more important than ozone photolysis in polluted environments (Mao et al., 2010;Tan et al., 2019). Ozonolysis of alkenes, isoprene, and monoterpenes could be important radical source in forest areas (Griffith et al., 2013). A discussion on this missing radical source and its impact would be helpful.

We have revised our calculation of  $P(HO_x)$  to include photolysis of HONO when measurements of HONO are available as well as alkene ozonolysis. The resulting  $P(HO_x)$  rates and radical concentrations did not significantly change, suggesting that these radical sources are not large contributors to the radical budget in our dataset. We have revised the manuscript to include these new radical sources in our calculations:

**Page 5:** "When these radicals were not available, OH, and  $HO_2$  radical concentrations were also calculated iteratively based on the total rate of  $HO_x$  radical production by  $O_3$  photolysis, HCHO photolysis, and alkene ozonolysis. When HONO was measured, HONO photolysis was also included as an OH source."

**Equation A7:**

$$P(\text{OH}) = \frac{2j_{\text{O}_3 \to \text{O}^1\text{D}}[\text{O}_3] \cdot k_{\text{O}^1\text{D} + \text{H}_2\text{O}}[\text{H}_2\text{O}]}{k_{\text{O}^1\text{D} + \text{H}_2\text{O}}[\text{H}_2\text{O}] + k_{\text{O}^1\text{D} + M}[M]} + j_{\text{HONO}}[\text{HONO}] + k_{\text{HO}_2 + \text{NO}}[\text{HO}_2][\text{NO}] + k_{\text{O}_3 + \text{RH}}[\text{O}_3][\text{RH}]Y_{\text{OH}}$$

Furthermore, based on comments from all 3 reviewers, we have added a sensitivity analysis of how the use of modeled radical concentrations affects our results:

**Page 16:** "In order to test the accuracy of the modeling, we used periods when HO2, OH, and NO were all measured and calculated how the production ratio  $P(\text{RONO}_2)/P(\text{HNO}_3)$  changed if modeled radical concentrations were used instead. These results are shown in Fig. A1. Even in the worst-case scenario (modeled concentrations used for all radicals), the slope is close to one (Fig. A1a), indicating that the use of modeled radicals does not significantly affect our results. Furthermore, Fig A1b–d show that the use of modeled OH or HO2 concentrations alone does not lead to noticeable changes in  $P(\text{RONO}_2)/P(\text{HNO}_3)$ . Use of modeled NO concentrations can cause small but noticeable changes in  $P(\text{RONO}_2)/P(\text{HNO}_3)$ , but modeled NO concentrations are used in less than 3% of all data points used in this analysis (238 out of 7988 data points)."

Figure A1: "Comparison of  $P(\text{RONO}_2)/P(\text{HNO}_3)$  when measured concentrations of all possible radicals are used (*x*-axis) versus when measured concentrations are replaced by modeled concentrations (*y*-axis). Panel a shows the result when modeled concentrations of OH, HO2, and NO are all used simultaneously; Panels b–d show the effect of replacing measured with modeled values one species at a time."

Also, I assume these radical concentrations are used to calculate the P(HNO3) and P(RONO2) separately for each 13 campaigns but it's not clear in the context. The authors should make this point clearer:

Yes, the radical concentrations were calculated separately for each

campaign and used to calculate  $P(HNO_3)$  and  $P(RONO_2)$ :

**Page 5:** "Where available, measurements of OH and HO2 were used to directly calculate RO2 formation and loss. When these radicals were not available, OH, and HO2 radical concentrations were also calculated iteratively based on the total rate of HOx radical production by O3 photolysis, HCHO photolysis, and alkene ozonolysis. When HONO was measured, HONO photolysis was also included as an OH source. In a small fraction of cases (3% of all data points), NO measurement are not available and NO concentration were calculated based on the concentrations of O3 and NO2. Details of the radical modeling, including the equations used to calculate the production and loss of these radicals, are given in Appendix A. "

(2) The detail description of alpha. The organic nitrate yield is determined for different VOCs and explained in the appendix. However, I would suggest adding a table to describe the range of organic nitrate yield. As I found in Figure 2, the least- squares fit is  $y=(1+125x^{-1.06})^{-1}$  and 125 should be equal to 1/(alpha\*fNO\*YRO2), the alpha would be 0.008 if fNO and YRO2 are unity. In this case, a comparison with least-squares fit to Eq. 4 and discussion on the parameters would be helpful to the reader to understand what the meaning of such fit function is.

We have added an additional figure to the manuscript showing values of alpha, fNO, and YRO2 as a function of NO2R/VOCR, as well as a paragraph describing the variation in these three parameters:

**Page 8:** "While the fraction of NOx loss occurring via RONO2 chemistry can be well predicted from just the NO2R/VOCR ratio, the observations exhibit a sharper transition from HNO3-dominated to RONO2-dominated NOx loss than would be expected if the VOC mixture remained constant. This effect can be explained by variation in  $Y_{RO_{2}eff}$ ,  $\alpha_{eff}$ , and  $f_{NO_{eff}}$  as NO2R/VOCR changes. The behavior of these three parameters is shown in Fig. 4. As NO2R/VOCR decreases,  $f_{NO_{eff}}$  consistently decreases from 0.8 to 0.2, due almost entirely to the decrease in NOx concentrations. In contrast, both  $Y_{RO_{2}eff}$  and  $\alpha_{eff}$  are larger in areas with low NO2R/VOCR ratios, due to changes in the VOC mixture between environments. In areas where NO2R/VOCR is high, many of the predominant VOCs, including CO, HCHO, and aromatics, either do not produce RO2 radicals when oxidized by OH or produce  $RO_2$  radicals that do not efficiently produce organic nitrates, leading to the relatively low values of both these parameters. In areas with low NO2R/VOCR ratios, the VOC mixture is often dominated by biogenic alkenes such as isoprene and monoterpenes that efficiently produce organic nitrates, leading to higher values of both  $Y_{RO_{2eff}}$  and  $\alpha_{eff}$ . However, although variation in these parameters can help explain some of the observed behavior of fractional NOx loss, the overall variation is much smaller than the variation of the NO2R/VOCR ratio. Each of the three parameters varies by a factor of 4 or less, while the NO2R/VOCR ratio varies by a factor of 1000."

**Figure 4:** "VOC oxidation parameters ( $\alpha_{eff}$ ,  $f_{NO_{eff}}$ ,  $Y_{RO_{2eff}}$ ) as a function of NO2R/VOCR. Used data points are restricted to the continental summer daytime boundary layer (i.e., over land, less than 1.5 km above ground level, and average temperature > 10 °C). The line and solid shapes show the median in each bin, and the vertical lines show an example of the interquartile range for each binned parameter."

(3) Some argumentation are too general or without explanation.

Page 4 Line 23, please define low-NOx

Because the terms  $low-NO_x$  and  $high-NO_x$  do not have an agreed upon definition and often cause confusion, we have removed these terms from this manuscript:

**Page 4:** " $f_{NO_i}$  is close to 1 under polluted or moderately-polluted conditions, but decreases as the concentration of NOx decreases."

Page 9 Line 1-2, according to Fig. 3, this statement relies on an assumption that many regions are located in the transition regime (e.g. NO2R/VOCR ranges from 2e-2 to 1e-3). The authors should provide relevant information to support this argument.

We have added a histogram of showing the number of observations as a function of NO2R/VOCR:

**Page 5:** "Although these field campaigns do not constitute a random sample of the atmosphere, the combined dataset provides an excellent survey of atmospheric chemistry over a wide range of conditions. The combined dataset includes nearly 8000 data points for which fractional NOx loss can be

calculated, spanning nearly 3 orders of magnitude in the ratio of NO2R to VOCR with no significant gaps (Fig. 2.)"

**Figure 2:** "Number of points in each bin for which the fraction of NOx loss occurring via RONO2 chemistry could be calculated."

**Page 11:** "Given the large number of data points sampled in this transition regime (Fig. 2), many regions of the United States are therefore likely to transition from a regime where  $HNO_3$  dominates  $NO_x$  loss to a mixed or  $RONO_2$ -dominated regime."

Page 9 Line 12, please define comprehensive metric.

We have reworded the sentence to clarify:

**Page 11:** "When considering ozone pollution on regional scales, OPE is a more appropriate metric than instantaneous ozone production because it accounts for ozone production both locally and further afield. "

*Page 5 Line 2, please provide the reaction rate constant for OH+NO2 reaction and the literature*

We have added this information to Appendix A:

**Page 15:** "The reaction rate constant for  $NO_2$  with OH was taken from Mollner et al. (2010), with temperature- and pressure-dependencies from Henderson et al. (2012)."

Technical corrections: Page 5 Line 1, 'NO2R' 2 should be subscripted. Page 9 Line 10, 'P(O3)' 3 should be subscripted. Page 13 Line 14, (A3) 2jHCHO\*[HCHO] Page 13 Line 19, (A8) L(OH) should be (VOCR+NO2R)\*[OH]

We thank the reviewer for bringing these errors to our attention

**Response to Reviewer 3**

This referee agrees with what already suggested by the other reviewers. In particular the point regarding the calculation of the OH, HO2 and RO2 and radicals should be better discussed. As pointed out a better analysis of the uncertainties for the calculation should be done together with the inclusion, if possible, of HONO photolysis and ozonolysis of unsaturated compounds. The calculation, as is at the moment, is very simplified (for example, why not including reaction with CO when considering the losses of OH radicals?) and it can well be that it is good enough for this study but a sensitivity check by adding additional sources would help understand their impact. In addition, the comparison, where possible, with the available radical measurements would also help understanding the reliability of the simple calculation used.

We have revised our calculation of  $P(HO_x)$  to include photolysis of HONO when measurements of HONO are available as well as alkene ozonolysis. The resulting  $P(HO_x)$  rates and radical concentrations did not significantly change, suggesting that these radical sources are not large contributors to the radical budget in our dataset. We have revised the manuscript to include these new radical sources in our calculations:

**Page 5:** "When these radicals were not available, OH, and  $HO_2$  radical concentrations were also calculated iteratively based on the total rate of  $HO_x$  radical production by  $O_3$  photolysis, HCHO photolysis, and alkene ozonolysis. When HONO was measured, HONO photolysis was also included as an OH source."

**Equation A7:**

$$P(\text{OH}) = \frac{2j_{\text{O}_3 \to \text{O}^1\text{D}}[\text{O}_3] \cdot k_{\text{O}^1\text{D}+\text{H}_2\text{O}}[\text{H}_2\text{O}]}{k_{\text{O}^1\text{D}+\text{H}_2\text{O}}[\text{H}_2\text{O}] + k_{\text{O}^1\text{D}+\text{M}}[M]} + j_{\text{HONO}}[\text{HONO}] + k_{\text{HO}_2+\text{NO}}[\text{HO}_2][\text{NO}] + k_{\text{O}_3+\text{RH}}[\text{O}_3][\text{RH}]Y_{\text{OH}}$$

Furthermore, based on comments from all 3 reviewers, we have added a sensitivity analysis of how the use of modeled radical concentrations affects our results:

Page 16: "In order to test the accuracy of the modeling, we used periods

when HO2, OH, and NO were all measured and calculated how the production ratio  $P(\text{RONO}_2)/P(\text{HNO}_3)$  changed if modeled radical concentrations were used instead. These results are shown in Fig. A1. Even in the worst-case scenario (modeled concentrations used for all radicals), the slope is close to one (Fig. A1a), indicating that the use of modeled radicals does not significantly affect our results. Furthermore, Fig A1b–d show that the use of modeled OH or HO2 concentrations alone does not lead to noticeable changes in  $P(\text{RONO}_2)/P(\text{HNO}_3)$ . Use of modeled NO concentrations can cause small but noticeable changes in  $P(\text{RONO}_2)/P(\text{HNO}_3)$ , but modeled NO concentrations are used in less than 3% of all data points used in this analysis (238 out of 7988 data points)."

**Figure A1**: "Comparison of  $P(\text{RONO}_2)/P(\text{HNO}_3)$  when measured concentrations of all possible radicals are used (*x*-axis) versus when measured concentrations are replaced by modeled concentrations (*y*-axis). Panel a shows the result when modeled concentrations of OH, HO2, and NO are all used simultaneously; Panels b–d show the effect of replacing measured with modeled values one species at a time."

Finally, we have revised our explanation of OH reactivity to clarify that loss of OH by reaction with CO is included, as well as the reaction of OH with other compounds 
[revised manuscript text omitted]

$$\quad \mathrm{NO} + \mathrm{O}_3 \to \mathrm{NO}_2 + \mathrm{O}_2 \tag{R1}$$

$$NO_2 + h\nu + O_2 \rightarrow O_3 + NO \tag{R2}$$

$$OH + R + O_2 \rightarrow RO_2 + H_2O \tag{R3}$$

$$\operatorname{RO}_2 + \operatorname{NO} \xrightarrow{1-\alpha} \operatorname{RO} + \operatorname{NO}_2$$
 (R4a)

$$RO + O_2 \rightarrow R'CHO + HO_2$$
 (R5)

5

$$\mathrm{HO}_2 + \mathrm{NO} \to \mathrm{OH} + \mathrm{NO}_2$$
 (R6)

The reactions that propagate the catalytic cycle occur at the same time as reactions that remove NOx from the atmosphere, terminating the cycle. Direct HNO3 production occurs through the association of OH with NO2 (R7). RONO2 compounds are produced as a minor channel of the RO2 + NO reaction (R4b). Some fraction of the time  $\alpha$ , these two radicals will associate to form an organic nitrate, with the balance forming NO2 and eventually producing O2 (R4a). The branching ratio

10

associate to form an organic nitrate, with the balance forming NO2 and eventually producing O3 (R4a). The branching ratio  $k_{R4b}/(k_{R4a} + k_{R4b})$  is designated  $\alpha$  and is determined by the nature of the R group as well as the temperature and pressure. Longer carbon backbones and lower temperatures increase  $\alpha$ , while lower pressures and oxygenated functional groups decrease it (Wennberg et al., 2018). Typical values of  $\alpha$  in the summertime continental boundary layer range from near 0 for small hydrocarbons and highly oxygenated compounds to over 0.20 for large alkanes and alkenes (Perring et al., 2013).

15
$$OH + NO_2 + M \rightarrow HNO_3 + M$$
 (R7)

$$\operatorname{RO}_2 + \operatorname{NO} + M \xrightarrow{\alpha} \operatorname{RONO}_2 + M$$
 (R4b)

The total rate of RONO2 production can be calculated from the properties of individual VOCs measured in the atmosphere via Eq. (1). In Eq. (1),  $Y_{RO_{2i}}$  represents the yield of RO2 radicals from VOC oxidation and  $f_{NO_i}$  represents the fraction of those RO2 radicals that react with NO instead of reacting with HO2 or undergoing unimolecular isomerization (e.g., Teng et al., 2017).  $f_{NO_i}$  is close to 1 under polluted or moderately-polluted conditions, but decreases in low-NOx conditionsas the concentration of NOx decreases.

$$P(\text{RONO}_2) = [\text{OH}] \sum_{R_i} [\text{R}_i] \cdot k_{\text{OH}+\text{R}_i} \cdot Y_{\text{RO}_{2_i}} \cdot f_{\text{NO}_i} \cdot \alpha_i$$
(1)

25 If the contributions from individual VOCs are summed and averaged, the total production of  $RONO_2$  can also be calculated from the effective behavior of the VOC mixture via Eq. (2), where VOCR is the sum of all measured VOC concentrations weighted by their reaction rate with OH.

$$P(\text{RONO}_2) = [\text{OH}] \cdot \text{VOCR} \cdot Y_{\text{RO}_{2\text{eff}}} \cdot f_{\text{NO}_{\text{eff}}} \cdot \alpha_{\text{eff}}$$
(2)

In a similar fashion, the production of HNO3 can be calculated via Eq. (3), where NO2R is the NO2 reactivity, or the concentration of NO2 multiplied by  $k_{\text{OH}+\text{NO}_2}$ . At 298 K and 1 atm, 10 ppb of NO2 is equivalent to an NO2R of 2.3 s-1.

$$P(\text{HNO}_3) = [\text{OH}] \cdot [\text{NO}_2] \cdot k_{\text{OH}+\text{NO}_2} = [\text{OH}] \cdot \text{NO2R}$$
(3)

Total  $NO_x$  loss is the sum of the conversion to  $HNO_3$  and conversion to  $RONO_2$ . The fraction of  $NO_x$  loss via  $RONO_2$ 5 production can be expressed analytically as Eq. 4.

$$\frac{P(\text{RONO}_2)}{P(\text{RONO}_2) + P(\text{HNO}_3)} = \left(1 + \frac{1}{\alpha_{\text{eff}} \cdot f_{\text{NO}_{\text{eff}}} \cdot Y_{\text{RO}_{2\text{eff}}}} \times \frac{\text{NO2R}}{\text{VOCR}}\right)^{-1} \tag{4}$$

The relative production of RONO2 and HNO3 is seen to be controlled by two factors, the first describing the chemistry of RO2 radicals ( $\alpha_{\text{eff}}$ ,  $f_{\text{NO}_{\text{eff}}}$ ,  $Y_{\text{RO}_{2\text{eff}}}$ ), and the second the ratio of NO2R to VOCR, which describes whether OH is more likely to react with a VOC or with NO2. Because Eq. 4 concerns fractional loss of NOx, the concentration of OH, which affects RONO2 and HNO3 production equally, does not appear in the result.

We show below that in the summertime continental boundary layer, the terms describing  $RO_2$  radical chemistry vary significantly less than the NO2R/VOCR ratio, allowing the relative importance of  $RONO_2$  and  $HNO_3$  chemistry to be roughly estimated from only a single ratio.

**3 Observed contributions of HNO3 and RONO2 chemistry to NOx loss**

**15 3.1 Daytime chemistry**

10

Relative RONO2 and HNO3 production rates were calculated for 13 separate campaign deployments in the northern hemisphere over the past 20 years. Campaigns were selected that included measurements of NOx, HNO3, O3, HCHO, a wide range of VOCs, and total organic nitrates ( $\Sigma$ RONO2). Although they do not include measurements of  $\Sigma$ RONO2, ITCT2k2 and CALNEX-P3 were also included to provide a pair of measurements of VOCs and NOx in the same geographic location

- 20 separated in time. A list of all campaigns used in this study is given in Table 1. Where available, measurements of OH and HO2 were used to directly calculate RO2 formation and loss; when . When these radicals were not available, OH,  $\frac{\text{HO}_2}{\text{HO}_2}$ , and  $\frac{\text{RO}_2}{\text{And HO}_2}$  radical concentrations were also calculated iteratively based on the total rate of HOx radical production by O3 and HCHO photolysis. Equations photolysis, HCHO photolysis, and alkene ozonolysis. When HONO was measured, HONO photolysis was also included as an OH source. In a small fraction of cases (3% of all data points), NO measurement are not
- 25 available and NO concentration were calculated based on the concentrations of  $O_3$  and  $NO_2$ . Details of the radical modeling, including the equations used to calculate the production and loss of these radicals, are given in Appendix A.

Although these field campaigns do not constitute a random sample of the atmosphere, the combined dataset provides an excellent survey of atmospheric chemistry over a wide range of conditions. The combined dataset includes nearly 8000 data points for which fractional NOx loss can be calculated, spanning nearly 3 orders of magnitude in the ratio of NO2R to VOCR

30 with no significant gaps (Fig. 2).

| Campaign name | Data Reference                 | Format   | Year | Base of Operations      | Date            |
|---------------|--------------------------------|----------|------|-------------------------|-----------------|
| ITCT2k2       | ITCT Science Team (2002)       | Airborne | 2002 | Monterey, CA            | 22 Apr – 19 May |
| INTEX-NA      | INTEX-A Science Team (2006)    | Airborne | 2004 | Palmdale, CA            | 2 Jul           |
|               |                                |          |      | Mascoutah, IL           | 7 Jul – 14 Jul  |
|               |                                |          |      | Portsmouth, NH          | 16 Jul – 10 Aug |
|               |                                |          |      | Mascoutah, IL           | 12 Aug          |
| INTEX-B       | INTEX-B Science Team (2011)    | Airborne | 2006 | Houston, TX             | 4 Mar – 19 Mar  |
|               |                                |          |      | Honolulu, HI            | 23 Apr – 28 Apr |
|               |                                |          |      | Anchorage, AK           | 1 May – 12 May  |
| BEARPEX 2007  | BEAPREX 07 Science Team (2007) | Ground   | 2007 | Georgetown, CA          | 15 Aug -10 Oct  |
| ARCTAS-B      | ARCTAS-B Science Team (2011)   | Airborne | 2008 | Palmdale, CA            | 18 Jun – 24 Jun |
|               |                                |          |      | Cold Lake, Alberta, CAN | 29 Jun – 8 Jul  |
|               |                                |          |      | Thule, Greenland        | 8 Jul – 10 Jul  |
| BEARPEX 2009  | BEAPREX 09 Science Team (2009) | Ground   | 2009 | Georgetown, CA          | 15 Jun – 31 Jul |
| CALNEX-P3     | CALNEX Science Team (2002a)    | Airborne | 2010 | Ontario, CA             | 1 May – 22 Jun  |
| CALNEX-SJV    | CALNEX Science Team (2002b)    | Ground   | 2010 | Bakersfield, CA         | 15 May – 30 Jun |
| DC3           | DC3 Science Team (2013)        | Airborne | 2012 | Salina, KS              | 13 May – 30 Jun |
| SOAS          | SOAS Science Team (2013)       | Ground   | 2013 | Centreville, AL         | 1 Jun – 15 Jul  |
| SEAC4RS       | SEAC4RS Science Team (2014)    | Airborne | 2013 | Houston, TX             | 8 Aug – 23 Sep  |
| FRAPPÉ        | FRAPPÉ Science Team (2014)     | Airborne | 2014 | Broomfield, CO          | 16 Jul –16 Aug  |
| KORUS-AQ      | KORUS-AQ Science Team (2018)   | Airborne | 2016 | Pyeongtaek, ROK         | 1 May – 14 Jun  |
|               |                                |          |      | Palmdale, CA            | 17 Jun – 18 Jun |

**Table 1. Field campaigns used in this analysis**

10

The fraction of total NOx loss occurring via RONO2 chemistry from all 13 of these campaigns is shown in Fig. 3a for points within the continental summertime boundary layer. Despite spanning a large range of environments, all 13 campaigns are well described by a single function of the form  $(1 + b \cdot (\frac{\text{NO2R}}{\text{VOCR}})^m)^{-1}$  (red line in Fig. 3a). This roughly matches the expected form functional form corresponds to a linear relationship between  $P(\text{RONO}_2)/P(\text{HNO}_3)$  and NO2R/VOCR on a log-log scale.

5 If *m* is fixed to 1, then this form also corresponds to the expected behavior if the VOC mixture were constant did not change between environments, and so all parameters other than NO2R/VOCR remained constant (gray line in Fig. 3a). However, the observations exhibit a sharper transition from HNO3-dominated to RONO2-dominated NOx loss, likely due to an increase in  $\alpha_{\text{eff}}$  as NO2R/VOCR decreases.

The calculated increase in fractional  $NO_x$  loss via  $RONO_2$  chemistry as NO2R/VOCR decreases is matched by an increase in the observed ratio of  $\Sigma RONO_2$  to the sum of  $\Sigma RONO_2$  and  $HNO_3$  (Fig. 3b). However, the increase in fractional concentrations as NO2R/VOCR decreases is much less than the increase in fractional production. At low NO2R/VOCR ratios, the

Figure 2. Comparison Number of the relative production rates of RONO2 and HNO3 as a function of NO2R/VOCR. Used data points are restricted to in each bin for which the continental summertime boundary layer (i.e., over land, less than 1.5 km above ground level, and average temperature > 10 °C). The top panel shows the fraction of NOx loss attributable to occurring via RONO2 chemistry , as well as a least-squares fit to the data and the expected behavior if  $\alpha_{eff}$ ,  $f_{NO_{eff}}$ ,  $Y_{RO_{2eff}}$  were constant could be calculated. The bottom panel shows the ratio of  $\Sigma RONO_2$  to the sum of HNO3 and  $\Sigma RONO_2$ . In each panel, the blue diamonds show the median in each bin and the vertical lines show the interquartile range.

dominant  $RONO_2$  species are typically short lived and can undergo heterogeneous hydrolysis to produce  $HNO_3$  (e.g., Browne et al., 2013). This indirect source of  $HNO_3$  can be the greatest source of  $HNO_3$  in forested environments, and leads to the relatively weak dependence of fractional concentration on NO2R/VOCR.

While the fraction of NOx loss occurring via RONO2 chemistry can be well predicted from just the NO2R/VOCR ratio,

- 5 the observations exhibit a sharper transition from HNO3-dominated to RONO2-dominated NOx loss than would be expected if the VOC mixture remained constant. This effect can be explained by variation in  $Y_{RO_{2eff}}$ ,  $\alpha_{eff}$ , and  $f_{NO_{eff}}$  as NO2R/VOCR changes. The behavior of these three parameters is shown in Fig. 4. As NO2R/VOCR decreases,  $f_{NO_{eff}}$  consistently decreases from 0.8 to 0.2, due almost entirely to the decrease in NOx concentrations. In contrast, both  $Y_{RO_{2eff}}$  and  $\alpha_{eff}$  are larger in areas with low NO2R/VOCR ratios, due to changes in the VOC mixture between environments. In areas where NO2R/VOCR
- 10 is high, many of the predominant VOCs, including CO, HCHO, and aromatics, either do not produce  $RO_2$  radicals when oxidized by OH or produce  $RO_2$  radicals that do not efficiently produce organic nitrates, leading to the relatively low values of both these parameters. In areas with low NO2R/VOCR ratios, the VOC mixture is often dominated by biogenic alkenes such as isoprene and monoterpenes that efficiently produce organic nitrates, leading to higher values of both  $Y_{RO_{2eff}}$  and  $\alpha_{eff}$ . However, although variation in these parameters can help explain some of the observed behavior of fractional NOx loss, the
- 15 overall variation is much smaller than the variation of the NO2R/VOCR ratio. Each of the three parameters varies by a factor of 4 or less, while the NO2R/VOCR ratio varies by a factor of 1000.

The conclusion that variation in VOC parameters is small compared to the variation in the NO2R/VOCR ratio does not hold outside of the summertime continental boundary layer. In the remote marine boundary layer or in the upper troposphere,  $\alpha_{eff}$